# Dual-Flow: Transferable Multi-Target, Instance-Agnostic Attacks via *In-the-wild* Cascading Flow Optimization

**Yixiao Chen**[1,*], **Shikun Sun**[1,*], **Jianshu Li**[2], **Ruoyu Li**[2], **Zhe Li**[2], **Junliang Xing**[1]

[1]Tsinghua University, [2]Ant Group

{chenyixi22, ssk21}@mails.tsinghua.edu.cn,
{jianshu.l, ruoyu.li, lizhe.lz}@antgroup.com, jlxing@tsinghua.edu.cn

## Abstract

Adversarial attacks are widely used to evaluate model robustness, and in black-box scenarios, the transferability of these attacks becomes crucial. Existing generator-based attacks have excellent generalization and transferability due to their instance-agnostic nature. However, when training generators for multi-target tasks, the success rate of transfer attacks is relatively low due to the limitations of the model's capacity. To address these challenges, we propose a novel Dual-Flow framework for multi-target instance-agnostic adversarial attacks, utilizing Cascading Distribution Shift Training to develop an adversarial velocity function. Extensive experiments demonstrate that Dual-Flow significantly improves transferability over previous multi-target generative attacks. For example, it increases the success rate from Inception-v3 to ResNet-152 by 34.58%. Furthermore, our attack method shows substantially stronger robustness against defense mechanisms, such as adversarially trained models. The code of Dual-Flow is available at: https://github.com/Chyxx/Dual-Flow.

## 1 Introduction

Deep neural networks (DNNs) are highly vulnerable to adversarial attacks [53, 22, 3, 19], which can significantly compromise their reliability. Among these, targeted black-box attacks—where adversaries manipulate a model into misclassifying an input as a specific target class without direct access to the model—are the most challenging and impactful [1, 2, 39].

Adversarial attacks can be classified into instance-specific and instance-agnostic approaches. Instance-specific attacks [12, 67, 16] optimize perturbations for each input image using victim model gradients but often suffer from poor transferability. Instance-agnostic attacks [65, 42, 69, 17] generalize perturbations over the dataset, leading to stronger black-box transferability. These methods typically rely on universal adversarial perturbations [41, 70] or generative models [46, 43].

Generative model-based attacks can be further divided into single-target [42, 18, 61] and multi-target [69, 17] approaches. While single-target attacks achieve high success rates, they require training a separate model per target class, making them impractical for large-scale attacks. Multi-target attacks address this by conditioning a single generator on target labels but often suffer from reduced transferability and weak robustness against adversarial defenses.

Diffusion models [29, 51, 48] offer strong generative capabilities, making them a promising tool for adversarial attacks. However, current diffusion-based attacks are all instance-specific methods, which

---

[*]Equal contribution; the order of co-first authors is interchangeable.

39th Conference on Neural Information Processing Systems (NeurIPS 2025).

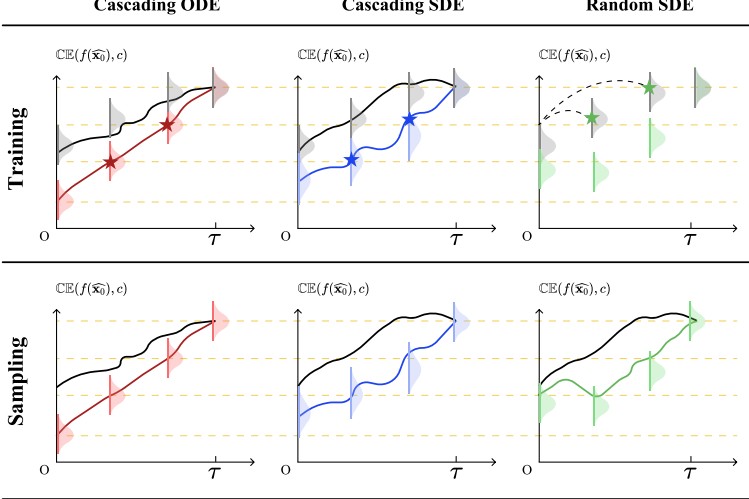

Figure 1: The comparison between Cascading ODE, Cascading SDE, and Random SDE for the second flow. The star shape represents the input for training the reverse flow. Notably, the Random SDE is observed to optimize in an incorrect distribution.

means they need to access the target model's gradient information during inference. Moreover, choosing between stochastic and deterministic sampling methods for adversarial perturbation generation remains an open challenge.

To address these challenges, we propose a novel **Dual-Flow framework** for multi-target instance-agnostic adversarial attacks. Our approach integrates (1) a pretrained diffusion model to generate an intermediate perturbation distribution as the forward flow and (2) a fine-tuned lightweight LoRA-based velocity function as reverse flow for targeted adversarial refinement. We introduce **Cascading Distribution Shift Training** to improve attack capability and employ **dynamic gradient clipping** to enforce the $\ell_\infty$ constraint.

As illustrated by the Cascading ODE and Cascading SDE in Figure 1, we first follow the black trajectory to introduce a slight perturbation to the image. Then, we follow either the red or blue trajectory to generate an altered image, effectively exploiting this process to attack the target model. Our main contributions are:

- **First application of flow-based ODE velocity training for adversarial attacks**, extending diffusion-based techniques beyond conventional score function training.

- **Dual-Flow algorithm**, integrating a pretrained diffusion-based forward ODE with a fine-tuned adversarial velocity function for structured perturbation generation.

- **Theoretical contribution on cascading improvement mechanism**, demonstrating how our method facilitates cascading improvements at later timesteps.

Extensive experiments demonstrate that our attack method achieves state-of-the-art black-box transferability in multi-target scenarios and exhibits high robustness against defense mechanisms.

## 2 Preliminary

### 2.1 Instance-Agnostic Attacks

Instance-agnostic attacks [39, 33, 60, 46, 69, 43, 42] learn perturbations based on data distributions rather than individual instances. These approaches, employing universal adversarial perturbations[41, 70] or generative models, have demonstrated superior transferability. This paper primarily focuses on the latter due to its greater flexibility and attack effectiveness.

Early generative model-based methods were primarily single-target attacks[65, 42, 43, 18, 61], requiring a separate model to be trained for each target class. Although these models exhibited high attack capabilities, the excessive training overhead limited their applicability when many target classes needed to be attacked. Recent research has proposed several multi-target attack methods[69, 17] that condition the perturbation generative model on class labels[69] or text embeddings[17] of classes. These approaches allow a single model to be trained to attack multiple target classes, significantly reducing the training overhead.

Consider a white-box image classifier characterized by the parameter $\theta$, denoted as $f : \mathcal{X} \to \mathcal{Y}$, where the input space $\mathcal{X} \subset \mathbb{R}^{C \times H \times W}$ corresponds to the image domain, and the output space $\mathcal{Y} \subset \mathbb{R}^L$ represents the confidence scores across various classes. Here, $L$ denotes the total number of classes. Given an original image $\mathbf{x} \in \mathcal{X}$ and a target class $c \in \mathcal{C}$, the goal of transferable multi-target generative attack is to generate the perturbation $\boldsymbol{\delta} = G(\mathbf{x}, c)$ and the pertubed image $\mathbf{x}^\epsilon = \mathbf{x} + \boldsymbol{\delta}$, in such a way that an unseen victim model $F$ predicts $c$ for the perturbed image, *i.e.*, $\arg\max_{i \in \mathcal{C}} F(\mathbf{x}^\epsilon)_i = c$. Here $G$ is the generator trained on the known source model $f_\theta$. To ensure that the manipulated images remain visually indistinguishable from the originals, the perturbation is constrained using the $l_\infty$ norm such that $\|\mathbf{x} - \mathbf{x}^\epsilon\|_\infty = \|\boldsymbol{\delta}\|_\infty < \epsilon$.

## 2.2 Diffusion Models and Flow-based Models

Diffusion models [50, 29, 51, 52] have emerged as powerful generative models, particularly for continuous data such as audio [32] and images [48]. Recently, Flow-based generative models [36, 15, 37], developed directly from ordinary differential equations, have also gained significant momentum. Given their strong generative capabilities, exploring their applications in adversarial attacks is natural.

**Sampling Algorithms.** One of the most appealing aspects of diffusion models is the flexibility in designing the sampling process. The generation process of diffusion models primarily follows two formulations: one based on the Stochastic Differential Equation (SDE) and the other on the Ordinary Differential Equation (ODE). Each approach has strengths and weaknesses, making them suitable for different scenarios.

**Diffusion Models in Adversarial Attack.** Currently, diffusion models have found some applications in adversarial attacks. Some utilize diffusion models to create unrestricted adversarial examples[5, 8], while others perform instance-specific attacks[68, 4]. However, they all rely on iterative optimization using the classifier model's gradients while generating adversarial examples, thus not qualifying as instance-agnostic attacks.

## 3 Dual-Flow for Adversarial Attack

We propose a Dual-Flow pipeline designed to transform an image $\mathbf{x} \in \mathcal{X}$ through a perturbed distribution $\mathcal{X}_\tau$ and ultimately into a constrained output space $\mathcal{X}^\epsilon$, which is $\{\mathbf{x}^\epsilon | \exists \mathbf{x} \in \mathcal{X}, \|\mathbf{x}^\epsilon - \mathbf{x}\|_\infty < \epsilon\}$. By construction, $\mathcal{X}^\epsilon$ enforces a maximum $\ell_\infty$ perturbation of $\epsilon$. Our method is instance-agnostic, which means that once training is completed, our model can generate adversarial examples during inference without requiring any access to the classifier model.

Specifically, we leverage the original ODE-based diffusion flow to map $\mathcal{X}$ to $\mathcal{X}_\tau$. Using a pretrained diffusion model's velocity function $\mathbf{v}_\phi(\cdot, \cdot)$ and a given input image $\mathbf{x} \sim \mathcal{X}$, we select a fixed timestep $\tau \in (0, 1)$. The perturbed image $\mathbf{x}_\tau \sim \mathcal{X}_\tau$ is obtained by integrating the following equation:

$$\frac{\partial}{\partial t}\Phi(\mathbf{x}, t) = \mathbf{v}_\phi(\Phi(\mathbf{x}, t), t), \quad \Phi(\mathbf{x}, 0) \sim \mathcal{X}, \tag{1}$$

from $t = 0$ to $t = \tau$.

To further map $\mathcal{X}_\tau$ to $\mathcal{X}^\epsilon$, we fine-tune a LoRA-based score function[30], yielding a new velocity function $\mathbf{v}_\theta$. Then by integrating another equation:

$$\frac{\partial}{\partial t}\Psi(\mathbf{x}, t) = \mathbf{v}_\theta(\Psi(\mathbf{x}, t), t), \quad \Psi(\mathbf{x}, \tau) \sim \mathcal{X}_\tau, \tag{2}$$

from $t = \tau$ to $t = 0$.

To train this second flow, we introduce a novel *Cascading Distribution Shift Training* strategy, which addresses the challenges posed by the inaccessibility of intermediate distributions during training.

Finally, to ensure that outputs remain within $\mathcal{X}^\epsilon$, we apply dynamic gradient stops during training, coupled with hard clipping operations at the final timestep. This approach allows for richer intermediate representations while maintaining the required perturbation bounds.

Details of our approach are provided in the following subsections.

### 3.1 A Construction of Better Extend Flow

Firstly, we construct an extended flow based on $j$, which is the negative value of the cross-entropy function:

$$j = -\mathbb{CE}(f(\mathbf{x}), c), \tag{3}$$

where $f$ is the source model and $c$ is the target label.

**Proposition 1** (Morse Flow Construction). *Under mild assumptions on the $\mathcal{X}^\epsilon$ and the function $j$, there exists $\epsilon > 0$, a unique smooth flow.*

$$\Phi : \mathcal{X}^\epsilon \times [0, \epsilon] \to \mathcal{X}^\epsilon, \tag{4}$$

*satisfying:*

$$\frac{d}{dt}\Phi(\mathbf{x}, t) = \mathbf{v}(\Phi(\mathbf{x}, t)),$$
$$\Phi(\mathbf{x}, 0) = \mathbf{x}, \tag{5}$$

*such that:*

1. *$\mathbf{v} = \alpha(\mathbf{x})\nabla_{\mathbf{X}} j(\mathbf{x})$ almost everywhere*

2. *$j(\Phi(\mathbf{x}, \epsilon)) \geq j(\mathbf{x})$ for all $x \in \mathcal{X}^\epsilon$, and $>$ holds almost everywhere if $j$ is not trivial*

3. *Each $\Phi(\cdot, t) : \mathcal{X}^\epsilon \to \mathcal{X}^\epsilon$ is a diffeomorphism*

A detailed proof is provided in Appendix A.1. Proposition 1 indicates we can find better extend flow to hack $j$ by $\nabla j$. In the following paragraphs, we will construct a concrete algorithm to realize it along with an existing flow.

### 3.2 Cascading Distribution Shift Training

Although the *in-the-wild* ODE trajectory is deterministic, obtaining exact intermediate samples remains a significant challenge, which poses difficulties for our approach. To address this issue, we propose the **Cascading Distribution Shift Training Algorithm**, specifically designed to enhance adversarial attack efficacy through two key mechanisms: (1) Enforcing a cascading hacking effect, where each perturbation step incrementally contributes to misleading the source model. (2) Ensuring the final perturbation follows the prescribed $\ell_\infty$ constraint. Further details are provided in Algorithm 1, where $l$ denotes the conditional input in real diffusion models.

Our proposed training framework not only circumvents the challenge of inaccessible intermediate timesteps but also offers additional advantages. Proposition 2 formalizes that our algorithm progressively refines a coarse-to-fine representation, thus effectively leveraging information from ahead timesteps. More concretely, due to the continuity of the ODE, our training algorithm enables a cascading optimization within an increasingly refined space.

**Theorem 2** (Cascading Improvement at Adjoint Timesteps). *Consider two consecutive timesteps $t, t - \delta$. Following Algorithm 1, when comparing the cases with and without updating $\theta$ at $t$, updating $\theta$ results in an equal or lower cross-entropy for $\widehat{\mathbf{x}_0}$ at $t - \delta$ when $\delta$ is sufficiently small and all functions are smooth.*

One crucial consideration is constraining the final result within $\mathcal{X}^\epsilon$. There are two primary methods to achieve this. The first approach incorporates the original ODE trajectory during model tuning,

---

**Algorithm 1** Cascading Distribution Shift Training

---

**Input:** $\tau = N\delta$, stepsize $\delta$, model param. $\phi, \theta$, source model $f$, target labels set $\mathcal{C}$, training dataset $\{\mathbf{I}^i\}_{i \in \mathcal{I}}$, learning rate $l_r$
Initialize $\theta = \phi$.
**repeat**
   **for** $i \in \mathcal{I}$ **do**
      get $\mathbf{x}_0 = \mathbf{I}^i$
      sample $c \sim \mathcal{C}$
      **for** $t = 1$ **to** $N$ **do**
         $\mathbf{x}_{t\delta} = \mathbf{x}_{(t-1)\delta} + \mathbf{v}_\phi(\mathbf{x}_{(t-1)\delta}, (t-1)\delta, \varnothing)\delta$
      **end for**
      **for** $t = N$ **to** $1$ **do**
         $\mathbf{x}_{(t-1)\delta} = \mathbf{x}_{t\delta} - \mathbf{v}_\theta(\mathbf{x}_{t\delta}, t\delta, c)\delta$
         $\widehat{\mathbf{x}_0} = \mathbf{x}_{t\delta} - \mathbf{v}_\theta(\mathbf{x}_{t\delta}, t\delta, c)t\delta$
         $\widehat{\mathbf{x}_0} = \text{clip}\,(\widehat{\mathbf{x}_0}, \mathbf{x} - \epsilon, \mathbf{x} + \epsilon)$
         $\theta = \theta - l_r \cdot \nabla_\theta(\mathbb{CE}(f(\widehat{\mathbf{x}_0}), c))$
      **end for**
   **end for**
**until** $\mathbf{v}_\theta$ convergence
**Return:** Dual-Flow $\{\mathbf{v}_\phi, \mathbf{v}_\theta\}$

---

---

**Algorithm 2** Dual-Flow Sampling

---

**Input:** $\tau = N\delta$, stepsize $\delta$, image $\mathbf{I}$, target label $c$, Dual-Flow $\{\mathbf{v}_\phi, \mathbf{v}_\theta\}$
$\mathbf{x} = \mathbf{I}$.
**for** $t = 1$ **to** $N$ **do**
   $\mathbf{x}_{t\delta} = \mathbf{x}_{(t-1)\delta} + \mathbf{v}_\phi(\mathbf{x}_{(t-1)\delta}, (t-1)\delta, \varnothing)\delta$
**end for**
**for** $t = N$ **to** $1$ **do**
   $\mathbf{x}_{(t-1)\delta} = \mathbf{x}_{t\delta} - \mathbf{v}_\theta(\mathbf{x}_{t\delta}, t\delta, c)\delta$
**end for**
$\mathbf{x}_0 = \text{clip}\,(\mathbf{x}_0, \mathbf{x} - \epsilon, \mathbf{x} + \epsilon)$
**Return:** $\mathbf{x}_0$

---

ensuring the output remains close to the original ODE flow. The second approach enforces the $\ell_\infty$ constraint or applies gradient clipping to suppress the influence of out-of-range image regions, guaranteeing that only in-range rewards contribute to model optimization. Our experiments show that dynamic gradient clipping yields the best performance among these methods.

Given the fixed model capacity, the Cascading Distribution Shift Training algorithm ensures greater consistency between the training and sampling processes, improving performance. This is visually illustrated in Figure 1.

### 3.3 Dual-Flow Sampling

During the sampling process, following the proposed training algorithm, a given image $\mathbf{x} \in \mathcal{X}$ is first mapped to $\mathbf{x}_\tau$ via Eq. (1). Subsequently, it is transformed into an intermediate state $\mathbf{x}_0'$ using Eq. (2). Finally, a hard truncation is applied to obtain the qualified perturbed sample $\mathbf{x}_0 \in \mathcal{X}^\epsilon$.

### 3.4 Deterministic Flow vs. Stochastic Flow

An important consideration is the choice between a deterministic flow, modeled by an ODE, and a stochastic flow, modeled by an SDE. This decision primarily depends on the second flow, as the first is inherited from a pretrained diffusion model.

When translating the distribution $\mathcal{X}^\tau$ to $\mathcal{X}^\epsilon$, a simple rescaling and noise injection into $\mathcal{X}^\epsilon$ is insufficient to fully recover $\mathcal{X}^\tau$. Consequently, this transformation falls outside the standard diffusion-based framework.

Table 1: Attack success rates (%) for multi-target attacks on normally trained models using the ImageNet NeurIPS validation set. The perturbation budget is constrained to $l_\infty \leq 16/255$. * indicates white-box attacks. The results are averaged across 8 different target classes, and the overall average on the far right is computed solely for black-box attacks.

| Source | Method | Inc-v3 | Inc-v4 | Inc-Res-v2 | Res-152 | DN-121 | GoogleNet | VGG-16 | Average |
|--------|--------|--------|--------|-----------|---------|--------|-----------|--------|---------|
| Inc-v3 | MIM | 99.90* | 0.80 | 1.00 | 0.40 | 0.20 | 0.20 | 0.30 | 0.48 |
| | TI-MIM | 98.50* | 0.50 | 0.50 | 0.30 | 0.20 | 0.40 | 0.40 | 0.38 |
| | SI-MIM | 99.80* | 1.50 | 2.00 | 0.80 | 0.70 | 0.70 | 0.50 | 1.03 |
| | DIM | 95.60* | 2.70 | 0.50 | 0.80 | 1.10 | 0.40 | 0.80 | 1.05 |
| | TI-DIM | 96.00* | 1.10 | 1.20 | 0.50 | 0.50 | 0.50 | 0.80 | 0.77 |
| | SI-DIM | 90.20* | 3.80 | 4.40 | 2.00 | 2.20 | 1.70 | 1.40 | 2.58 |
| | Logit | 99.60* | 5.60 | 6.50 | 1.70 | 3.00 | 0.80 | 1.50 | 3.18 |
| | SU | 99.59* | 5.80 | 7.00 | 3.35 | 3.50 | 2.00 | 3.94 | 4.26 |
| | C-GSP | 93.40* | 66.90 | 66.60 | 41.60 | 46.40 | 40.00 | 45.00 | 51.08 |
| | CGNC | 96.03* | 59.43 | 48.06 | 42.48 | 62.98 | 51.33 | 52.54 | 52.80 |
| | Dual-Flow | 90.08* | **77.19** | **66.76** | **77.06** | **82.64** | **73.01** | **67.09** | **73.96** |
| Res-152 | MIM | 0.50 | 0.40 | 0.60 | 99.70* | 0.30 | 0.30 | 0.20 | 0.38 |
| | TI-MIM | 0.30 | 0.30 | 0.30 | 96.50* | 0.30 | 0.40 | 0.30 | 0.32 |
| | SI-MIM | 1.30 | 1.20 | 1.60 | 99.50* | 1.00 | 1.40 | 0.70 | 1.20 |
| | DIM | 2.30 | 2.20 | 3.00 | 92.30* | 0.20 | 0.80 | 0.70 | 1.53 |
| | TI-DIM | 0.80 | 0.70 | 1.00 | 90.60* | 0.60 | 0.80 | 0.50 | 0.73 |
| | SI-DIM | 4.20 | 4.80 | 5.40 | 90.50* | 4.20 | 3.60 | 2.00 | 4.03 |
| | Logit | 10.10 | 10.70 | 12.80 | 95.70* | 12.70 | 3.70 | 9.20 | 9.87 |
| | SU | 12.36 | 11.31 | 16.16 | 95.08* | 16.13 | 6.55 | 14.28 | 12.80 |
| | C-GSP | 37.70 | 47.60 | 45.10 | 93.20* | 64.20 | 41.70 | 45.90 | 47.03 |
| | CGNC | 53.39 | 51.53 | 34.24 | 95.85* | 85.66 | 62.23 | 63.36 | 58.40 |
| | Dual-Flow | **69.58** | **71.92** | **56.10** | 92.39* | **85.73** | **73.65** | **67.59** | **70.76** |

A natural solution is to construct an *in-the-wild* ODE (as we do) or SDE that maps $\mathcal{X}^\tau$ to $\mathcal{X}^\epsilon$. In the ODE formulation, we define a velocity function $\mathbf{v}_\theta$ that satisfies Eq. (2). A similar approach applies to the SDE case, where stochastic noise facilitates the distribution transition.

When comparing from the perspectives of randomness and determinism, as shown in Figure 1, we label our framework as **Cascading ODE** and implement two SDE-based training algorithms. One injects noise at a random timestep within $(0, \tau)$, labeled as **Random SDE**, while the other first directly adds noise at $\tau$ and then reverses the flow using DDPM, resulting in a weak cascading relationship, labeled as **Cascading SDE**. While SDE-based training algorithms more closely resemble original diffusion models, they present two key challenges.

First, **Cascading SDE** introduces a random term, which may make it more difficult to construct the Cascading Improvement relationship as stated in Proposition 2 for **Cascading ODE**. Second, sampling with SDEs tends to produce unstable results, where larger step sizes exacerbate accumulated errors, further impacting reliability, as demonstrated in our experiments.

As for **Random SDE**, it exhibits the worst performance because when sampling $\mathbf{x}_t$ by directly adding noise, the distribution of $\mathbf{x}_t$ remains unchanged. Consequently, even slight training of the reverse flow leads to a distribution mismatch, as illustrated by the star in the last column of Figure 1.

## 4 Experiments

### 4.1 Experimental Settings

**Dataset.** Following [69, 18, 17], we train the model on the ImageNet training set[10] and evaluate the attack performance using ImageNet-NeurIPS (1k) dataset proposed by NeurIPS 2017 adversarial competition[44].

**Victim Models.** We consider various naturally trained models, including Inception-v3 (Inc-v3) [55], Inception-v4 (Inc-v4) [56], Inception-ResNet-v2 (Inc-Res-v2) [56], ResNet-152 (Res-152) [27], DenseNet-121 (DN-121) [31], GoogleNet [54], and VGG-16 [49].

Table 2: Attack success rates (%) for single-target attacks against normally trained models on ImageNet NeurIPS validation set. Note that CGNC$^\dagger$ and Dual-Flow$^\dagger$ denote the single-target variants of CGNC and our proposed Dual-Flow, respectively. The perturbation budget is constrained to $l_\infty \leq 16/255$. * indicates white-box attacks. The results are averaged across 8 different target classes, and the overall average on the far right is computed solely for black-box attacks.

| Source | Method | Inc-v3 | Inc-v4 | Inc-Res-v2 | Res-152 | DN-121 | GoogleNet | VGG-16 | Average |
|--------|--------|--------|--------|------------|---------|--------|-----------|--------|---------|
| Inc-v3 | GAP | 86.90* | 45.06 | 34.48 | 34.48 | 41.74 | 26.89 | 34.34 | 36.16 |
| | CD-AP | 94.20* | 57.60 | 60.10 | 37.10 | 41.60 | 32.30 | 41.70 | 45.07 |
| | TTP | 91.37* | 46.04 | 39.37 | 16.40 | 33.47 | 25.80 | 25.73 | 31.14 |
| | DGTA-PI | 94.63* | 67.95 | 55.03 | 50.50 | 47.38 | 47.67 | 48.11 | 52.77 |
| | CGNC$^\dagger$ | 98.84* | 74.76 | 64.48 | 62.00 | 78.94 | 69.06 | 70.74 | 70.00 |
| | Dual-Flow | 90.08* | 77.19 | 66.76 | 77.06 | 82.64 | 73.01 | 67.09 | 73.96 |
| | Dual-Flow$^\dagger$ | 91.41* | **78.85** | **70.59** | **79.12** | **83.36** | **77.52** | **71.29** | **76.79** |
| Res-152 | GAP | 30.99 | 31.43 | 20.48 | 84.86* | 58.35 | 29.89 | 39.70 | 35.14 |
| | CD-AP | 33.30 | 43.70 | 42.70 | 96.60* | 53.80 | 36.60 | 34.10 | 40.70 |
| | TTP | 62.03 | 49.20 | 38.70 | 95.12* | 82.96 | 65.09 | 62.82 | 60.13 |
| | DGTA-PI | 66.83 | 53.62 | 47.61 | 96.48* | 86.61 | 68.29 | 69.58 | 65.42 |
| | CGNC$^\dagger$ | 68.86 | 69.45 | 45.71 | 98.61* | **91.14** | 69.83 | 68.05 | 68.84 |
| | Dual-Flow | 69.58 | 71.92 | 56.10 | 92.39* | 85.73 | 73.65 | 67.59 | 70.76 |
| | Dual-Flow$^\dagger$ | **72.25** | **74.35** | **58.44** | 93.65* | 87.61 | **75.45** | **71.11** | **76.12** |

Table 3: Attack success rates (%) for multi-target attacks against robust models on ImageNet NeurIPS validation set. The perturbation budget $l_\infty \leq 16/255$. The results are averaged on 8 different target classes.

| Source | Method | Inc-v3$_{adv}$ | IR-v2$_{ens}$ | Res50$_{SIN}$ | Res50$_{IN}$ | Res50$_{fine}$ | Res50$_{Aug}$ | Average |
|--------|--------|----------------|---------------|---------------|--------------|----------------|---------------|---------|
| Inc-v3 | MIM | 0.16 | 0.10 | 0.20 | 0.27 | 0.44 | 0.19 | 0.23 |
| | TI-MIM | 0.21 | 0.19 | 0.33 | 0.49 | 0.68 | 0.31 | 0.37 |
| | SI-MIM | 0.13 | 0.19 | 0.26 | 0.43 | 0.63 | 0.29 | 0.32 |
| | DIM | 0.11 | 0.09 | 0.16 | 0.33 | 0.39 | 0.19 | 0.21 |
| | TI-DIM | 0.15 | 0.13 | 0.16 | 0.21 | 0.33 | 0.14 | 0.19 |
| | SI-DIM | 0.19 | 0.21 | 0.43 | 0.71 | 0.84 | 0.46 | 0.47 |
| | Logit | 0.30 | 0.30 | 0.70 | 1.23 | 3.14 | 0.86 | 1.09 |
| | SU | 0.49 | 0.41 | 0.84 | 1.75 | 3.55 | 1.04 | 1.35 |
| | C-GSP | 20.41 | 18.04 | 6.96 | 33.76 | 44.56 | 21.95 | 24.28 |
| | CGNC | 24.36 | 22.54 | 8.85 | 40.83 | 52.18 | 22.85 | 28.60 |
| | Ours | **51.54** | **55.62** | **45.86** | **74.56** | **78.54** | **67.56** | **62.28** |
| Res-152 | MIM | 0.19 | 0.15 | 0.28 | 1.58 | 2.75 | 0.78 | 0.96 |
| | TI-MIM | 0.61 | 0.73 | 0.50 | 2.51 | 4.75 | 1.76 | 1.81 |
| | SI-MIM | 0.24 | 0.24 | 0.39 | 0.66 | 0.84 | 0.36 | 0.46 |
| | DIM | 0.63 | 0.37 | 0.94 | 8.50 | 14.22 | 3.77 | 4.74 |
| | TI-DIM | 0.23 | 0.30 | 0.28 | 0.76 | 1.49 | 0.49 | 0.59 |
| | SI-DIM | 0.71 | 0.71 | 0.75 | 2.73 | 3.89 | 1.37 | 1.69 |
| | Logit | 1.15 | 1.18 | 1.65 | 6.70 | 15.46 | 5.93 | 5.34 |
| | SU | 2.12 | 1.20 | 1.95 | 7.53 | 21.14 | 6.95 | 6.82 |
| | C-GSP | 14.60 | 16.01 | 16.84 | 60.30 | 65.51 | 42.88 | 36.02 |
| | CGNC | 22.21 | 26.71 | 29.83 | 79.80 | 84.05 | 63.75 | 51.06 |
| | Ours | **44.50** | **54.09** | **59.35** | **83.05** | **84.28** | **76.35** | **66.94** |

For further evaluation, we also analyze the performance of our method on robustly trained models. These include adv-Inception-v3 (Inc-v3$_{adv}$) [22], ens-adv-Inception-ResNet-v2 (IR-v2$_{ens}$) [26], and several robustly trained ResNet-50 models. The ResNet-50 variants are: Res50$_{SIN}$ (trained on stylized ImageNet), Res50$_{IN}$ (trained on a mixture of stylized and Nature ImageNet), Res50$_{fine}$ (further fine-tuned with an auxiliary dataset [21]), and Res50$_{Aug}$ (trained with advanced data augmentation techniques from Augmix [28]).

**Baseline Methods.** We compare our attack with several attack methods. For instance-specific attacks, we consider MIM [12], DIM [66], SIM [35], DIM [13], Logit [71], and SU [62]. For instance-agnostic attacks, we consider C-GSP [69], CGNC [17], GAP [46], CD-AP [42], TTP [43], and DGTA-PI[18]. Among them, C-GSP [69]and CGNC [17]are multi-target generative attacks, and the others are single-target generative attacks. For SU attack [62], we choose to compare with its best version DTMI-Logit-SU. For CGNC [17], we also consider its single target variant and compare it to a single target attack method.

**Implementation Details.** We adopt stable-diffusion [48] as our pre-trained diffusion model. We set $\tau = 0.25$ and $N = 6$ for training and testing. The LoRA rank is 16.

Following previous work [69, 18, 17], we choose Res-152 and Inc-v3 as source models to train our model. The perturbation budget $\epsilon$ is 16/255. We conduct 50k steps of training for multi-target tasks. To compare our method with other single-target attacks, we further fine-tune our model for an additional 10k steps to specialize in a single target class(more details provided in Appendix D.2). For multi-target training, we use a learning rate of $2.5 \times 10^{-5}$ and a total batch size of 8 (distributed across two NVIDIA RTX 3090 GPUs, each with 24GB memory and batch size 4). Training under this setting takes approximately one day to complete. For single-target fine-tuning, we set the learning rate to $1 \times 10^{-5}$ and a batch size of 4, conducted on a single NVIDIA RTX 3090 GPU, which requires approximately 4 hours. In total, the experiments involve approximately 160 GPU hours.

## 4.2 Transferability Evaluation

We assess the effectiveness of our proposed Dual-Flow for black-box target attacks through a series of experiments. To ensure consistency with previous work [69, 18, 17], we select eight distinct target classes [70] to conduct the target black-box attack testing protocol. We use the average attack success rate (ASR) across 8 target classes as an evaluation metric.

**Multi-Target Black-Box Attack.** We initially conduct attacks on normally trained models to evaluate the performance of multi-target attacks. The results in Table 1 show that our proposed Dual-Flow method exhibits significantly superior transferability, outperforming state-of-the-art instance-specific and instance-agnostic methods. Specifically, our method achieves an average ASR improvement of 21.16% and 12.36% over CGNC [17] using Inc-v3 and Res-152 as source models, respectively, on black-box models. Notably, instance-specific methods, despite higher success rates in white-box settings, tend to overfit the source models' classification boundaries, resulting in poor performance when transferred to black-box models.

**Single-Target Black-Box Attack.** To further evaluate our method's effectiveness, we compare it with other state-of-the-art instance-agnostic single-target attacks. Multi-target attacks are inherently more challenging than single-target ones, disadvantaging our model in such comparisons. To ensure fairness, we applied a masked fine-tuning technique similar to CGNC [17], allowing us to fine-tune our model separately for each target class and create single-target variants.

The results in Table 2 show that after fine-tuning, Dual-Flow† achieves higher attack success rates and generally outperforms leading single-target methods. Notably, our method excels in average black-box attack capability even without individual fine-tuning for the eight target classes. This demonstrates our approach's significant capacity and effectiveness in multi-target attacks, reducing the need for separate models for each target class in resource-constrained scenarios.

## 4.3 Attack Under Defense Strategies

To demonstrate the robustness of our proposed Dual-Flow, we evaluate its performance against several widely used defense mechanisms.

**Robustly Trained Networks.** We first consider attacking six robustly trained networks, with results in Table 3. Attacking robustness-augmented models is challenging, as previous methods see a significant drop in success rates. However, our approach consistently misleads black-box classifiers into predicting the specified classes, showing marked improvement over earlier multi-target methods. Notably, using Inc-v3 as the source model, the average attack success rate against the six robust models increases significantly from 28.60% to 62.28%, highlighting our method's effectiveness.

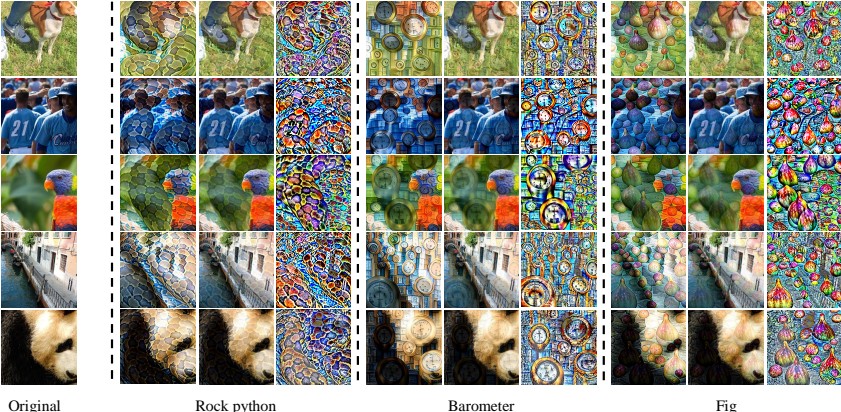

| Original | Rock python | Barometer | Fig |

Figure 2: Visualization results of different input images targeting various classes. For each text prompt of the target class, the left column displays the adversarial examples generated before clipping, the middle column shows the adversarial examples after clipping, and the right column presents the corresponding adversarial perturbations, which represent the differences between the clipped adversarial examples and the original images. Note that the perturbations are scaled to a range between 0 and 1. The source model used is Inc-v3.

**Input Process Defense.** We compared our method's performance with the state-of-the-art multi-target attack method CGNC against input preprocessing defenses, such as image smoothing [11] and JPEG compression [14]. As shown in Figure 3, our method consistently outperforms CGNC under these defenses. For example, using Inc-v3 as the source model and DN-121 as the target model, our method achieves a 52.99% success rate under Gaussian smoothing, compared to CGNC's 28.27%. This highlights the superior effectiveness of our approach in overcoming input preprocessing defenses.

### 4.4 Visualization

To gain a deeper understanding of the effectiveness of our method, we visualized both the unclipped and clipped samples generated by our approach. Additionally, for consistency with other perturbation-based attack methods, we visualized

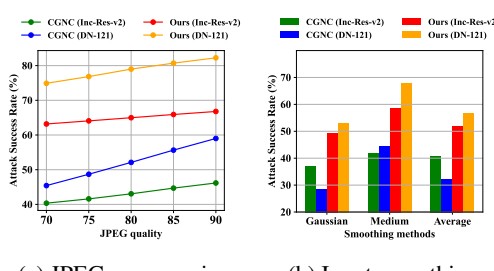

(a) JPEG compression     (b) Input smoothing

Figure 3: A comparison of CGNC and our method regarding attack success rates against various input processing defense strategies. The results against JPEG compression are shown in (a), while (b) presents the outcomes against different input smoothing methods. Inc-v3 is the source model and Inc-Res-v2, along with DN-121, are the target models.

the equivalent adversarial perturbations, defined as the pixel differences between the clipped samples and the clean images. As illustrated in Figure 2, our method first transforms the original image into one that maintains a similar layout and color scheme but becomes semantically closer to the target class. This transformed image is subsequently clipped to ensure its pixel differences from the original image remain within the epsilon bound. Visually, the clipped image retains substantial semantic features of the target class. Notably, the adversarial perturbations also exhibit distinct semantic patterns aligned with the target class, further validating the effectiveness of our approach.

### 4.5 Ablation Study

To further validate the effectiveness of our chosen Cascading ODE, we conducted a series of ablation experiments in this section. Here, Dual-Flow-co represents the original method, Dual-Flow-cs denotes the Cascading SDE variant, and Dual-Flow-rs denotes the Random SDE variant. During the reverse process at inference time, these variants em-

ploy either the DDPM scheduler or the DDIM scheduler. As shown in Table 4, our method significantly outperforms the other variants in white-box and black-box transfer attacks.

Furthermore, we compared the impact of different sampling steps $N$ during the reverse process. As shown in Figure 4, increasing the sampling steps steadily increases success rates for both Dual-Flow-co and Dual-Flow-cs. However, for Dual-Flow-rs, the success rate quickly declines as the inference steps increase, supporting our analyses in Section 3.4 and Figure 1.

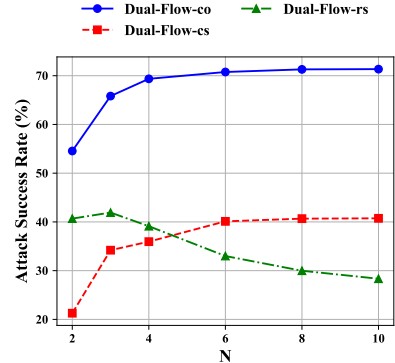

## 5  Discussion

**Potential Advantage of SDE.**  Although our experiments with Cascading SDE have not yet surpassed the performance of Cascading ODE, we believe that methods based on Schrödinger bridges [9] have the potential to bring significant improvements. Schrödinger bridge formulations

Figure 4: The multi-target black-box attack success rates of several variants of our method. The source model used is Res-152.

provide a principled way to learn stochastic transport maps, which could offer better control over the reverse trajectory and enhance the stability of the cascading distribution shift.

Table 4: The multi-target attack success rates of several variants of our method. The source model used is Res-152. The white-box attack success rate refers to the performance on Res-152, while the black-box attack success rate represents the average performance across six black-box models.

| Method | White-box | Black-box |
|---|---|---|
| Dual-Flow-co | 92.39 | 70.76 |
| Dual-Flow-cs + DDIM | 68.56 | 40.12 |
| Dual-Flow-cs + DDPM | 74.58 | 46.04 |
| Dual-Flow-rs + DDIM | 55.39 | 33.00 |
| Dual-Flow-rs + DDPM | 29.19 | 14.86 |

## 6  Conclusion

We have introduced Dual-Flow, a novel framework for highly transferable multi-target adversarial attacks. By employing Cascading Distribution Shift Training to develop an adversarial velocity function, our approach addresses the limitations of existing methods. Extensive experimental results demonstrate that Dual-Flow achieves remarkable improvements in transferability and robustness compared to previous multi-target generative attacks. These findings highlight the potential of Dual-Flow as a powerful tool for evaluating and improving the robustness of generative models.

## Acknowledgments and Disclosure of Funding

This work was supported in part by the Natural Science Foundation of China under Grant No. 62222606 and the Ant Group Security and Risk Management Fund. We sincerely thank Qixin Wang for her assistance in creating the figures for this paper.

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

# A  Proofs

## A.1  Proof of Morse Flow Construction

**Proposition 3** (Morse Flow Construction). *Let $B \subset \mathbb{R}^n$ be a bounded open set with smooth boundary, and let $j : B \to \mathbb{R}$ be a smooth Morse function that extends to $C^\infty(\overline{B})$. There exists $\varepsilon > 0$, a smooth vector field $X \in \mathfrak{X}(B)$, and a unique smooth flow*

$$\Phi : B \times [0, \varepsilon] \to B$$

*satisfying:*

$$\frac{d}{dt}\Phi(x, t) = X(\Phi(x, t)),$$
$$\Phi(x, 0) = x,$$

*such that:*

1. *$j(\Phi(x, \varepsilon)) \geq j(x)$ for all $x \in B$*

2. *Each $\Phi(\cdot, t) : B \to B$ is a diffeomorphism*

3. *Trajectories remain bounded away from $\partial B$ for $t \in [0, \varepsilon]$*

*Constructive Proof.* We proceed through coordinated geometric and analytic constructions.

**Step 1: Geometric Preparations**

1. *Smooth Defining Function*: By the smooth boundary assumption, there exists $\mu \in C^\infty(\overline{B}, [0, \infty))$ with:

   - $\mu^{-1}(0) = \partial B$
   - $\nabla\mu(x) \neq 0$ for $x \in \partial B$
   - $\mu(x) \sim \text{dist}(x, \partial B)$ near $\partial B$

   For explicit construction, take $\mu(x) = f(\text{dist}(x, \partial B))$ where $f \in C^\infty([0, \infty))$ satisfies $f(r) = r$ near 0.

2. *Critical Point Isolation*: Since $j$ is Morse on compact $\overline{B}$, its critical points $\mathscr{C}(j) = \{p_1, \ldots, p_N\}$ are finite and non-degenerate. Choose pairwise disjoint neighborhoods $U_i \ni p_i$ with:
   $$\overline{U_i} \subset B \setminus \partial B \quad \text{and} \quad \overline{U_i} \cap \mathscr{C}(j) = \{p_i\}$$

**Step 2: Vector Field Construction**

1. *Partition of Unity*: Let $\{\rho_i\}_{i=1}^N$ be smooth functions with:

   $$\text{supp}(\rho_i) \subset U_i, \quad 0 \leq \rho_i \leq 1, \quad \sum_{i=1}^N \rho_i \leq 1$$

   Define the cutoff function:

   $$\eta(x) := 1 - \sum_{i=1}^N \rho_i(x)$$

   Note $\eta \equiv 0$ near critical points and $\eta \equiv 1$ outside $\bigcup U_i$.

2. *Decay Modulation*: Fix $m \geq n + 1$. Define the boundary decay factor:

   $$\mu_m(x) := \mu(x)^m$$

   This ensures sufficient regularity at $\partial B$.

3. *Synthesized Vector Field*: Define

$$X(x) := \eta(x)\mu_m(x)\nabla j(x)$$

This field vanishes at critical points and near $\partial B$.

**Step 3: Flow Analysis**

*Boundary Avoidance*: For $x \in B$, let $r(t) = \mu(\Phi(x,t))$. Compute:

$$\frac{dr}{dt} = \nabla\mu(\Phi) \cdot X(\Phi) = \eta(\Phi)\mu_m(\Phi)\nabla\mu(\Phi) \cdot \nabla j(\Phi)$$

Using $|\nabla\mu \cdot \nabla j| \leq C$ near $\partial B$:

$$\left|\frac{dr}{dt}\right| \leq C\eta(\Phi)\mu(\Phi)^{m+1} \leq Cr(t)^{m+1}$$

Solutions to $\dot{r} \leq Cr^{m+1}$ satisfy it will never reach 0 in finite time, establishing boundary avoidance.

**Step 4: Monotonicity & Diffeomorphism**

1. *Energy Gain*: Along trajectories:

$$\frac{d}{dt}j(\Phi(x,t)) = \nabla j(\Phi) \cdot X(\Phi) = \eta(\Phi)\mu_m(\Phi)\|\nabla j(\Phi)\|^2 \geq 0$$

Thus $j$ is non-decreasing, with strict increase except at critical points.

2. *Flow Diffeomorphisms*: The differential $D\Phi(x,t)$ satisfies:

$$\frac{d}{dt}D\Phi(x,t) = DX(\Phi(x,t))D\Phi(x,t)$$

Since $X$ is smooth with bounded derivatives on $\overline{B}$, Grönwall's inequality gives:

$$\|D\Phi(x,t)\| \leq \exp\left(\int_1^{1+\varepsilon} \|DX(\Phi(x,s))\|ds\right) < \infty$$

Thus $\Phi(\cdot,t)$ remains locally diffeomorphic, and properness follows from boundary avoidance.

**Step 5: Isotopy Synthesis**

The time-$\varepsilon$ map $\Phi(0,\varepsilon)$ provides the required isotopy through diffeomorphisms. $\qquad\square$

## A.2 Proof of Cascading Improvement at Adjoint Timesteps

**Proposition 4** (Cascading Improvement at Adjoint Timesteps). *Consider two consecutive timesteps $t, t - \delta$. Following Algorithm 1, when comparing the cases with and without updating $\theta$ at $t$, updating $\theta$ results in an equal or lower cross-entropy for $\widehat{\mathbf{x}}_0$ at $t - \delta$ when $\delta$ is sufficiently small and all functions are smooth.*

*Proof.* We want to show

$$\mathbb{CE}\big(f(\widehat{\mathbf{x}}_0^2), c\big) \leq \mathbb{CE}\big(f(\widehat{\mathbf{x}}_0^1), c\big)$$

for sufficiently small $\delta$ with the following statement:

$$\widehat{\mathbf{x}}_0^1 = \mathbf{x}_{t-\delta} - \mathbf{v}_\theta\big(\mathbf{x}_{t-\delta}, t - \delta\big)(t - \delta)$$

and

$$\widehat{\mathbf{x}}_0^2 = \mathbf{x}_{t-\delta} - \mathbf{v}_{\theta+\Delta\theta}\big(\mathbf{x}_{t-\delta}, t - \delta\big)(t - \delta),$$

where

$$\Delta\theta = -l_r \nabla_\theta\Big(\mathbb{CE}\big(f(\widehat{\mathbf{x}}_0^0), c\big)\Big), \quad \widehat{\mathbf{x}}_0^0 = \mathbf{x}_t - \mathbf{v}_\theta\big(\mathbf{x}_t, t\big)t.$$

**Step 1. Relating $\widehat{\mathbf{x}_0}^1$ and $\widehat{\mathbf{x}_0}^0$.** Since $\mathbf{x}_{t-\delta}$ is close to $\mathbf{x}_t$ for small $\delta$, the smoothness of $\mathbf{v}_\theta(\cdot,\cdot)$ implies

$$\|\widehat{\mathbf{x}_0}^1 - \widehat{\mathbf{x}_0}^0\| = \left\|\left[\mathbf{x}_{t-\delta} - \mathbf{v}_\theta(\mathbf{x}_{t-\delta}, t-\delta)\,(t-\delta)\right] - \left[\mathbf{x}_t - \mathbf{v}_\theta(\mathbf{x}_t, t)\,t\right]\right\|$$

can be made arbitrarily small by taking $\delta$ sufficiently small (and using continuity/Lipschitz arguments). Consequently,

$$\nabla_\theta \mathbb{CE}\big(f(\widehat{\mathbf{x}_0}^1), c\big) \quad \text{and} \quad \nabla_\theta \mathbb{CE}\big(f(\widehat{\mathbf{x}_0}^0), c\big)$$

are also close for small $\delta$.

**Step 2. First-order comparison at $t - \delta$.** By a first-order expansion of $\mathbf{v}_{\theta+\Delta\theta}$ around $\theta$ and the smoothness of $\mathbf{v}_\theta(\cdot,\cdot)$, we have

$$\mathbf{v}_{\theta+\Delta\theta}(\mathbf{x}_{t-\delta}, t-\delta) = \mathbf{v}_\theta(\mathbf{x}_{t-\delta}, t-\delta) + \nabla_\theta \mathbf{v}_\theta(\mathbf{x}_{t-\delta}, t-\delta)\,\Delta\theta + \mathcal{O}\big(\|\Delta\theta\|^2\big).$$

Hence,

$$\widehat{\mathbf{x}_0}^2 - \widehat{\mathbf{x}_0}^1 = -\left[\mathbf{v}_{\theta+\Delta\theta}(\mathbf{x}_{t-\delta}, t-\delta) - \mathbf{v}_\theta(\mathbf{x}_{t-\delta}, t-\delta)\right](t-\delta) \approx -(t-\delta)\,\nabla_\theta \mathbf{v}_\theta\big(\mathbf{x}_{t-\delta}, t-\delta\big)\,\Delta\theta.$$

**Step 3. Cross-entropy decrease.** Using the smoothness of the cross-entropy and another first-order expansion,

$$\mathbb{CE}\big(f(\widehat{\mathbf{x}_0}^2), c\big) - \mathbb{CE}\big(f(\widehat{\mathbf{x}_0}^1), c\big)$$
$$\approx \big\langle \nabla_{\widehat{\mathbf{x}_0}} \mathbb{CE}\big(f(\widehat{\mathbf{x}_0}^1), c\big),\ \widehat{\mathbf{x}_0}^2 - \widehat{\mathbf{x}_0}^1 \big\rangle + \mathcal{O}\big(\|\widehat{\mathbf{x}_0}^2 - \widehat{\mathbf{x}_0}^1\|^2\big)$$
$$\approx \big\langle \nabla_\theta \mathbb{CE}\big(f(\widehat{\mathbf{x}_0}^1), c\big),\ \Delta\theta \big\rangle + \mathcal{O}\big(\|\Delta\theta\|^2, \|\delta\|\big).$$

By definition of the gradient step $\Delta\theta = -l_r\,\nabla_\theta \mathbb{CE}\big(f(\widehat{\mathbf{x}_0}^0), c\big)$ and the fact that $\nabla_\theta \mathbb{CE}\big(f(\widehat{\mathbf{x}_0}^0), c\big)$ is close to $\nabla_\theta \mathbb{CE}\big(f(\widehat{\mathbf{x}_0}^1), c\big)$ for small $\delta$, the above inner product is non-positive up to higher-order (small) terms. Concretely,

$$\big\langle \nabla_\theta \mathbb{CE}\big(f(\widehat{\mathbf{x}_0}^1), c\big),\ -l_r\,\nabla_\theta \mathbb{CE}\big(f(\widehat{\mathbf{x}_0}^0), c\big) \big\rangle \leq 0$$

when $\delta$ is sufficiently small so that these gradients align (up to small errors). Therefore,

$$\mathbb{CE}\big(f(\widehat{\mathbf{x}_0}^2), c\big) \leq \mathbb{CE}\big(f(\widehat{\mathbf{x}_0}^1), c\big),$$

which completes the proof. $\qquad\qquad\square$

# B  Related Works

## B.1  Targeted and Untargeted Attacks

**Targeted Attacks.**  The objective of targeted attacks is to force the classifier to output a specified label. In other words, the attacker seeks to cause the model to produce incorrect classification results and aims for the result to be a specific target class. This type of attack is more hazardous due to its ability to manipulate the model's output precisely but is typically more challenging to execute.

**Untargeted Attacks.**  The goal of untargeted attacks is to make the classifier output any incorrect label. The attacker merely needs to mislead the model so that its classification result does not match the true label. Despite having lower requirements, untargeted attacks can still have severe consequences in certain situations.

## B.2  White-Box and Black-Box Attacks

**White-Box Attacks.**  White-box attacks assume that the attacker has complete access to the target model, including its architecture, parameters, and gradient information. Using this information, the attacker can generate efficient adversarial examples through iterative optimization methods.

**Black-Box Attacks.** Black-box attacks assume that the attacker does not have access to the internal information of the target model. A common method to implement black-box attacks is to utilize transferability, where adversarial examples are first generated against a known source model and then used to attack the unknown target model.

### B.3 Instance-Specific and Instance-Agnostic Attacks

**Instance-Specific Attacks.** Instance-specific attacks [12, 20, 16, 67, 59, 38, 34] and *instance-agnostic* generate adversarial perturbations for specific input samples. The attacker uses gradient information from the target model and iterative optimization algorithms to create minimal perturbations that achieve the attack on a given sample. Such attacks usually have high success rates on individual samples but lack generalization and transferability.

**Instance-Agnostic Attacks.** Instance-agnostic attacks [65, 39, 33, 42, 43, 18] do not target specific input samples but instead learn universal adversarial perturbations or generative functions based on data distribution. These attack methods have better generalization across different samples, thus exhibiting stronger transferability.

### B.4 Subcategories of Instance-Agnostic Attacks

Instance-agnostic attacks can be further subdivided into the following categories:

**Universal Adversarial Perturbations.** These methods learn a universal perturbation [41, 70] applicable to the entire dataset. The classifier can be misled by superimposing this perturbation on any input sample.

**Generative Models.** Generative attacks [46, 42] train a generator that, upon receiving an input sample, can produce specific adversarial perturbations. This approach often surpasses universal adversarial perturbations regarding flexibility and attack efficacy.

### B.5 Single-Target and Multi-Target Attacks

**Single-Target Attacks.** Single-target attacks train an individual generative model for each target class [42, 43, 18, 61]. Although these models achieve high success rates for single-target classes, the training cost becomes substantial when the number of target classes is large, thereby limiting practical usability.

**Multi-Target Attacks.** Multi-target attacks simultaneously train the attack capabilities for multiple target classes within a single model [24, 69, 17]. Class labels or text embeddings are typically used as conditional inputs to generate corresponding adversarial perturbations. This method significantly reduces training costs and enhances feasibility in real-world applications.

## C Comparison with Other Diffusion-Based Methods

Currently, diffusion models have been explored in several adversarial attack methods [5, 68, 6, 4, 8]. Among them, ACA [6] and DiffAttack [4] leverage DDIM inversion to obtain the latent space of diffusion models and optimize adversarial examples within this latent space. AdvDiffuser [5], DiffPGD [68], and AdvDiff [8] incorporate adversarial guidance during the reverse denoising process of diffusion models. Notably, although these methods are diffusion-based, they are all instance-specific attacks, requiring access to the target classifier's gradient information during inference for each input sample to perform the attack.

In contrast, our method, once trained, does not require any further information from the classifier, enabling more efficient generation of adversarial examples.

Furthermore, since these prior methods either only support untargeted attacks[5, 68, 6, 4] or focus on unrestricted adversarial examples[5, 6, 4, 8], their settings are not directly compatible with ours, making direct comparisons infeasible.

# D  Method Details

## D.1  Target Class Condition Representation

For each label $c$ in the target label set $\mathcal{C}$, we first obtain its class description and format it into a text condition using the template "a photo of a {class}"[47]. Subsequently, we utilize CLIP's text encoder to derive this textual input's embedding $\mathbf{e}$. Finally, this embedding is fed into our model via cross-attention mechanisms:

$$
\begin{aligned}
Q = \mathbf{z}W_Q, K = \mathbf{e}W_K, V = \mathbf{e}W_V, \\
Attention(Q, K, V) = softmax(\frac{QK^T}{\sqrt{d}}) \cdot V,
\end{aligned}
\tag{6}
$$

where $\mathbf{z} \in \mathbb{R}^{d_z}$ denotes the flattened intermediate features of the unet model, $W_Q \in \mathbb{R}^{d_z \times d}$, $W_K \in \mathbb{R}^{d_e \times d}$, $W_V \in \mathbb{R}^{d_e \times d}$ are learnable parameters.

By employing this approach, we can leverage the rich semantic priors associated with the target classes embedded in the pre-trained diffusion model, thereby facilitating a more effective training process.

---

**Algorithm 3** Single-Target Fine-Tuning Mechanism

---

**Input:** $\tau = N\delta$, stepsize $\delta$, model param. $\phi, \theta$, model $f$, target label $c$, training dataset $\{\mathbf{I}^i\}_{i \in \mathcal{I}}$, learning rate $l_r$
**repeat**
  **for** $i \in \mathcal{I}$ **do**
    get $\mathbf{x}_0 = \mathbf{I}^i$
    **for** $t = 1$ **to** $N$ **do**
      $\mathbf{x}_{t\delta} = \mathbf{x}_{(t-1)\delta} + \mathbf{v}_\phi(\mathbf{x}_{(t-1)\delta}, (t-1)\delta, \varnothing)\delta$
    **end for**
    **for** $t = N$ **to** $1$ **do**
      $\mathbf{x}_{(t-1)\delta} = \mathbf{x}_{t\delta} - \mathbf{v}_\theta(\mathbf{x}_{t\delta}, t\delta, c)\delta$
      $\widehat{\mathbf{x}_0} = \mathbf{x}_{t\delta} - \mathbf{v}_\theta(\mathbf{x}_{t\delta}, t\delta, c)t\delta$
      get random mask $M$
      $\widehat{\mathbf{x}_0} = \mathbf{x}_0 + M \cdot (\widehat{\mathbf{x}_0} - \mathbf{x}_0)$
      $\widehat{\mathbf{x}_0} = \text{clip}\,(\widehat{\mathbf{x}_0}, \mathbf{x} - \epsilon, \mathbf{x} + \epsilon)$
      $\theta = \theta - l_r \cdot \nabla_\theta(\mathbb{CE}(f(\widehat{\mathbf{x}_0}), c))$
    **end for**
  **end for**
**until** $\mathbf{v}_\theta$ convergence
**Return:** Dual-Flow $\{\mathbf{v}_\phi, \mathbf{v}_\theta\}$

---

## D.2  Fine-Tuning on Single-Target Tasks

We fine-tune our model for single-target tasks to enhance its performance further. Specifically, we fix the target label during training, enabling the model to focus on targeted attacks for a specific label. To mitigate the perturbations being confined to some areas of the image, which can reduce the robustness and transferability of adversarial examples in single-target training, we apply the mechanism introduced in [17].

In detail, we generate a random mask $M$ of the same size as the image, where several randomly positioned square pixel areas are set to 0, and the rest are set to 1. By multiplying this mask with the perturbation, we ensure the generated adversarial samples remain consistent with the original image in the masked square areas. This forces the model to create adversarial patterns distributed across the entire image rather than being localized to specific regions, as illustrated in Algorithm 3.

Like other single-target methods, we must fine-tune a separate model for each target. However, due to our model's powerful capabilities in multi-target attacks, once the model is trained on the

multi-target task, it requires only a few additional steps to adapt to each single-target task. This results in significantly lower training overhead compared to other methods.

# E   Computational Cost

For training the multi-target Dual-Flow (e.g., using Res-152 as the source model), we require approximately 24 hours of training on 2 NVIDIA RTX 3090 GPUs, which is equivalent to 48 GPU hours. To further fine-tune on single-target tasks, we need an additional 4 hours per class on a single NVIDIA RTX 3090 GPU, totaling 4 hours × 8 classes = 32 GPU hours. Once training is complete, our method only requires 15 minutes of inference on a single GPU to generate adversarial samples for 8 target classes across the entire ImageNet NeurIPS validation set.

Table 5: Training and inference time for Dual-Flow with Res-152 as the source model.

| Train (only multi-target) | Train (multi-target + single-target) | Inference |
|---|---|---|
| 48 GPU hours | 80 GPU hours | 15 min |

We highlight that once our method completes training, it can rapidly generate adversarial samples with high transferability and robustness without requiring any gradient information from the classifier models.

# F   More Experiments

**Evaluation on Transformer Models.**   We tested our attack method's success rate when transferred to transformer models. Specifically, we utilized Res152 as the source model. The results, included in Table 6, demonstrate that despite the fundamental architectural differences between transformer models and our source model based on convolutional networks (Res152), our method maintains a high attack success rate, significantly outperforming baseline methods. This further corroborates the advantage of our method in terms of transferability.

Table 6: Attack success rates (%) for multi-target attacks on transformer models. The source model is Res-152.

| Method | ViT-B/16 [64] | CaiT-S/24 [58] | Visformer-S [7] | DeiT-B [57] | LeViT-256 [23] | TNT-S [25] |
|---|---|---|---|---|---|---|
| C-GSP | 11.78 | 32.00 | 36.60 | 35.58 | 37.85 | 31.00 |
| CGNC | 19.46 | 54.56 | 58.70 | 59.90 | 57.53 | 48.40 |
| Dual-Flow | **36.39** | **74.24** | **76.72** | **78.50** | **79.34** | **67.86** |

**Evaluation on DiffPure.**   We evaluate our attack method using Diffusion Models for Adversarial Purification (DiffPure)[45]. The experimental results show the attack success rates of our method under various DiffPure $t^*$ settings and compare them with the baseline method. As illustrated in Table 7, the baseline method is easily nullified by the purification process, whereas our method maintains a significant success rate. This further demonstrates the robustness of our approach.

Table 7: Attack success rates (%) for multi-target attacks on normally trained models with DiffPure. The source model is Res-152.

| $t^*$ | Method | Inc-v3 | Inc-v4 | Inc-Res-v2 | Res-152 | DN-121 | GoogleNet | VGG-16 |
|---|---|---|---|---|---|---|---|---|
| 0.05 | CGNC | 16.26 | 19.91 | 8.53 | 67.76 | 49.81 | 21.79 | 29.81 |
|  | Dual-Flow | 49.60 | 51.92 | 37.56 | 79.50 | 70.20 | 51.30 | 52.26 |
| 0.10 | CGNC | 2.41 | 2.96 | 1.25 | 14.65 | 10.10 | 3.46 | 4.59 |
|  | Dual-Flow | 24.31 | 25.32 | 18.76 | 48.05 | 39.81 | 25.06 | 24.78 |
| 0.15 | CGNC | 0.47 | 0.46 | 0.34 | 1.84 | 1.46 | 0.62 | 0.92 |
|  | Dual-Flow | 7.20 | 7.87 | 5.89 | 16.70 | 13.72 | 7.71 | 8.34 |

Table 8: Attack success rates (%) for multi-target attacks on normally trained models using the ImageNet validation set. The perturbation budget is constrained to $l_\infty \le 16/255$. * indicates white-box attacks. The results are averaged across 8 different target classes, and the overall average on the far right is computed solely for black-box attacks.

| Source | Method | Inc-v3 | Inc-v4 | Inc-Res-v2 | Res-152 | DN-121 | GoogleNet | VGG-16 | Average |
|---|---|---|---|---|---|---|---|---|---|
| Inc-v3 | CGNC | 96.59* | 57.82 | 46.84 | 44.13 | 65.90 | 53.40 | 56.27 | 54.06 |
| | Dual-Flow | 89.89* | **75.74** | **65.05** | **75.73** | **82.75** | **72.21** | **66.20** | **72.95** |
| Res-152 | CGNC | 56.00 | 50.37 | 32.26 | 96.44* | 86.69 | 63.84 | 63.90 | 58.84 |
| | Dual-Flow | **69.75** | **72.53** | **54.11** | 92.70* | **86.71** | **74.08** | **68.22** | **70.90** |

**Transferability Evaluation On ImageNet Validation Set.** In addition to the evaluation on the ImageNet-NeurIPS (1k) dataset[44], we conducted an assessment of our attack method on the ImageNet validation set (50k)[10] and compared it with the state-of-the-art multi-target attack method, CGNC[17]. The experimental results presented in Table 8 indicate that our method achieves a significantly higher average black-box attack success rate than CGNC, demonstrating its superior transferability. This outcome is consistent with the results observed on the ImageNet-NeurIPS (1k) dataset.

**Evaluation on Smaller Perturbation Budgets.** To validate the effectiveness of our method when adversarial perturbations are more imperceptible, we conducted experiments at lower perturbation budgets for black-box targeted attacks. We used the $\epsilon = 16/255$ versions of trained models (CGNC and Dual-Flow) and clipped the generated samples to meet the $\epsilon = 12/255$ and $\epsilon = 8/255$ limits. The results are shown in Tab 9. The results show that our method achieves better attack performance than the baseline across various perturbation budgets.

Table 9: Attack success rates (%) for multi-target attacks on normally trained models with different perturbation budgets. The source model is Inc-v3.

| $\epsilon$ | Method | Inc-v4 | Inc-Res-v2 | Res-152 | DN-121 | GoogleNet | VGG-16 |
|---|---|---|---|---|---|---|---|
| 16/255 | CGNC | 59.43 | 48.06 | 42.48 | 62.98 | 51.33 | 52.54 |
| | Dual-Flow | **77.19** | **66.76** | **77.06** | **82.64** | **73.01** | **67.09** |
| 12/255 | CGNC | 43.23 | 30.34 | 28.48 | 46.96 | 33.65 | 37.95 |
| | Dual-Flow | **55.80** | **41.82** | **54.41** | **63.19** | **47.66** | **44.94** |
| 8/255 | CGNC | 13.33 | 6.54 | 6.95 | 14.65 | 7.51 | 10.29 |
| | Dual-Flow | **23.86** | **14.04** | **21.21** | **27.68** | **13.39** | **15.55** |

**Evaluation on Single-Target Attacks against Robustly Trained Models.** We validate the effectiveness of our attack method in single-target settings against the robust models mentioned in the main text and compare it with other single-target attacks. As shown in Table 10, the conclusions are similar to those in the multi-target case in the main text, demonstrating that our method can effectively mislead robustly trained classifiers.

Table 10: Attack success rates (%) for single-target attacks against robust models on ImageNet NeurIPS validation set. The perturbation budget $l_\infty \le 16/255$. The results are averaged on 8 different target classes. The source model is Res-152.

| Method | Inc-v3$_{adv}$ | IR-v2$_{ens}$ | Res50$_{SIN}$ | Res50$_{IN}$ | Res50$_{fine}$ | Res50$_{Aug}$ |
|---|---|---|---|---|---|---|
| GAP | 5.72 | 4.51 | 7.33 | 71.04 | 83.64 | 52.07 |
| CD-AP | 3.77 | 6.48 | 7.09 | 63.72 | 76.79 | 49.67 |
| TTP | 27.99 | 26.08 | 24.61 | 72.47 | 74.51 | 70.96 |
| DGTA-PI | 31.10 | 30.07 | 27.70 | 77.13 | 80.55 | 76.78 |
| CGNC$^\dagger$ | 31.55 | 33.63 | 33.31 | **88.34** | **89.74** | 72.96 |
| Dual-Flow$^\dagger$ | **47.32** | **56.22** | **60.66** | 84.15 | 85.18 | **78.56** |

# G   Victim Model Details

All the victim models we used employ the official weights provided for the 1K-class ImageNet dataset [10]. For normally trained models (including transformer models), we directly call these models and their weights through the torchvision [40] or timm [63] libraries. For robust models (adversarially trained models):

- Inc-v3$_{adv}$ and IR-v2$_{ens}$, we call their official weights through the timm library.

- Res50$_{SIN}$, Res50$_{IN}$, and Res50$_{fine}$, we use the weights provided by the open-source repository Texture-vs-Shape [21].

- Res50$_{Aug}$, we use the weights provided by the open-source repository AugMix [28].

We report the accuracy of these models on the ImageNet NeurIPS validation set in Tab 11.

Table 11: Accuracy (%) of victim models on ImageNet NeurIPS validation set.

| Type | Model & Accuracy | | | | | | |
|---|---|---|---|---|---|---|---|
| Normal | Inc-v3 | Inc-v4 | Inc-Res-v2 | Res-152 | DN-121 | GoogleNet | VGG-16 |
| | 95.1 | 94.7 | 97.3 | 94.5 | 90.7 | 87.0 | 87.2 |
| Robust | Inc-v3$_{adv}$ | IR-v2$_{ens}$ | Res50$_{SIN}$ | Res50$_{IN}$ | Res50$_{fine}$ | Res50$_{Aug}$ | |
| | 86.7 | 94.5 | 68.4 | 89.5 | 93.2 | 94.1 | |
| Transformer | ViT-B/16 | CaiT-S/24 | Visformer-S | DeiT-B | LeViT-256 | TNT-S | |
| | 84.4 | 98.1 | 96.5 | 96.9 | 94.7 | 91.3 | |

# H   More Ablation Studies

**Ablation on LoRA Fine-tuning.**   To demonstrate that the adversarial attack capability of our model stems from LoRA fine-tuning, we evaluated the model's performance when performing attacks directly without applying LoRA. As shown in Table 12, the original model fails to achieve effective adversarial attacks, which validates the necessity of incorporating LoRA for fine-tuning.

Table 12: Attack success rates comparing the model with and without LoRA. The source model is Res-152.

| Method | Inc-v3 | Inc-v4 | Inc-Res-v2 | Res-152 | DN-121 | GoogleNet | VGG-16 |
|---|---|---|---|---|---|---|---|
| w/o LoRA | 0.075 | 0.075 | 0.05 | 0.05 | 0.075 | 0.05 | 0.075 |
| w/ LoRA | 69.58 | 71.92 | 56.10 | 92.39 | 85.73 | 73.65 | 67.59 |

**Ablation on Dynamic Gradient Clipping.**   To evaluate the effect of dynamic gradient clipping, we designed a variant that does not apply dynamic clipping during training and only clips the outputs to satisfy the norm constraints during inference. As shown in Table XXX, if the model is not trained with clipping, it fails to adapt to the norm constraints at inference time and thus cannot perform effective attacks.

Table 13: Attack success rates comparing the model with and without Dynamic Gradient Clipping. The source model is Res-152.

| Method | Inc-v3 | Inc-v4 | Inc-Res-v2 | Res-152 | DN-121 | GoogleNet | VGG-16 |
|---|---|---|---|---|---|---|---|
| w/o clip | 1.85 | 2.32 | 1.96 | 2.84 | 3.51 | 1.82 | 1.19 |
| w/ clip | 69.58 | 71.92 | 56.10 | 92.39 | 85.73 | 73.65 | 67.59 |

**Ablation on Loss.** We designed a variant: during training, we use a new $L_2$ loss function to make the model's output as close as possible to the original ODE trajectory, ensuring it does not deviate too far from the original ODE flow. We call this variant Dual-Flow-$L_2$. The experimental results in Table 14 indicate that the attack capability of this variant is not ideal, as described in Section 3.2.

Table 14: Comparison of Dual-Flow and Dual-Flow-$L_2$. The source model is Res-152.

| Method | Inc-v3 | Inc-v4 | Inc-Res-v2 | Res-152 | DN-121 | GoogleNet | VGG-16 | Average |
|---|---|---|---|---|---|---|---|---|
| Dual-Flow | 69.58 | 71.92 | 56.10 | 92.39 | 85.73 | 73.65 | 67.59 | 70.76 |
| Dual-Flow-$L_2$ | 54.90 | 56.26 | 42.94 | 86.80 | 78.41 | 57.71 | 46.36 | 56.10 |

# I  More Analysis

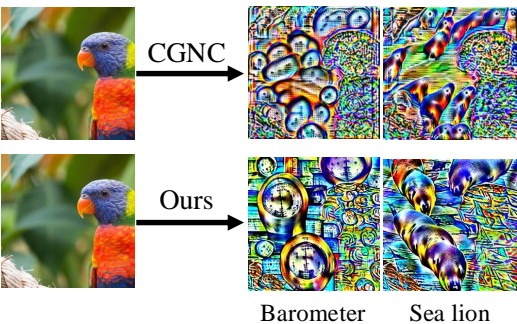

Figure 5: Visualization results comparing the adversarial perturbations generated by our method with those produced by CGNC.

Table 15: Comparison of different types of adversarial inputs. CGNC-P and Dual-Flow-P represent the adversarial perturbations generated by CGNC and our method, respectively, while Dual-Flow-A denotes the unclipped adversarial samples produced by our method. The adversarial perturbations are scaled to a range between 0 and 1 before input into the classifier.

| Source | Method | Inc-v3 | Inc-v4 | Inc-Res-v2 | Res-152 | DN-121 | GoogleNet | VGG-16 |
|---|---|---|---|---|---|---|---|---|
| Inc-v3 | CGNC-P | 56.80* | 19.15 | 22.06 | 10.56 | 13.14 | 14.56 | 3.41 |
| | Dual-Flow-P | 86.55* | 81.20 | 74.55 | 74.55 | 70.66 | 76.15 | 55.10 |
| | Dual-Flow-A | 99.12* | 95.31 | 92.79 | 97.39 | 96.80 | 95.62 | 87.95 |
| Res-152 | CGNC-P | 23.61 | 25.72 | 39.07 | 58.29* | 39.09 | 37.47 | 17.21 |
| | Dual-Flow-P | 68.44 | 81.48 | 78.26 | 86.29* | 80.44 | 74.59 | 55.35 |
| | Dual-Flow-A | 94.38 | 96.60 | 94.62 | 99.19* | 97.92 | 93.54 | 90.85 |

**Semantic Adversarial Attack.** We compared the adversarial perturbations generated by our method and those produced by the state-of-the-art multi-target attack method, CGNC[17]. As illustrated in Figure 5, the visual results indicate that while CGNC's perturbations contain some semantic features of the target class, they are primarily confined to small, repetitive patterns. In contrast, our method generates perturbations that are semantically more representative of the complete target class.

To further validate this observation, we directly input the adversarial perturbations generated by CGNC and our method into the target classifier. As shown in Table 15, our adversarial perturbations alone can induce the classifier to predict the target class with a relatively high probability. Conversely, the perturbations produced by CGNC exhibit a lower likelihood, particularly when transferred to black-box models. This demonstrates that our perturbations incorporate more semantic features of the target class.

Moreover, as depicted in Figure 2, the unclipped samples generated by our method are semantically very close to the target class. We also input these unclipped samples directly into the target classifier. Table 15 shows that these samples are highly likely to be classified as the target class. This confirms that semantic proximity to the target class effectively increases attack success rates.

These findings collectively suggest that our attack method's robustness and transferability result from embedding substantial target class semantics into the images, thereby reducing dependence on specific target model decision boundaries.

## J More Visualization

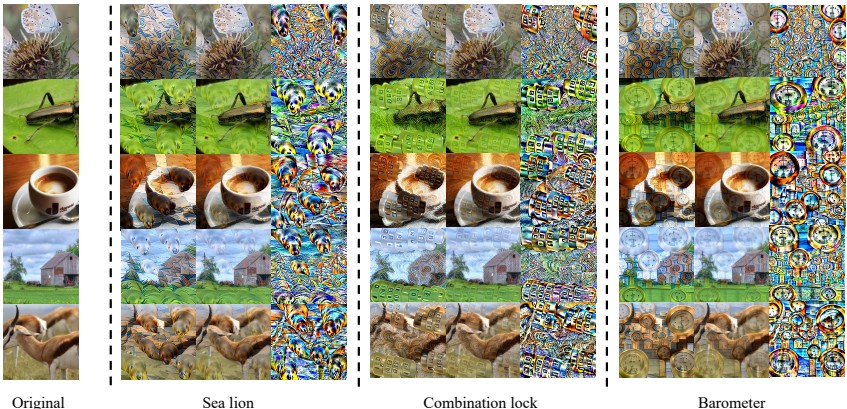

| Original | Sea lion | Combination lock | Barometer |

Figure 6: Visualization results of different input images targeting various classes. For each text prompt of the target class, the left column displays the adversarial examples generated before clipping, the middle column shows the adversarial examples after clipping, and the right column presents the corresponding adversarial perturbations, which represent the differences between the clipped adversarial examples and the original images. Note that the perturbations are scaled to a range between 0 and 1. The source model used is Inc-v3.

## K Limitations

While our proposed Dual-Flow framework demonstrates strong adversarial attack performance and transferability, several limitations remain. First, the training process involves additional computational overhead compared to some simpler attack methods, which may affect scalability to extremely large datasets or models. Second, although we observe improved semantic consistency in generated perturbations, there is still room to enhance the interpretability and controllability of adversarial patterns in more complex scenarios. Lastly, our evaluations focus primarily on standard image classification benchmarks; extending the approach to other tasks or modalities warrants further investigation.

## L Statistical Significance

To rigorously evaluate the reliability of our attack success rates, we partitioned the test dataset into 5 disjoint subsets, each containing 200 images. We computed the attack success rate for each subset independently and derived the overall 95% confidence intervals (CIs) for our method. These confidence intervals provide a quantitative measure of variability and statistical uncertainty. We further compared our results against the baseline method by examining the overlap of their respective confidence intervals. As shown in Table 16, the non-overlapping intervals observed in our experiments indicate that the improvement in attack success rate achieved by our method is statistically significant with high confidence.

Table 16: Attack success rates with 95% confidence intervals over 5 data splits, compared to the baseline method. The source model is Res-152.

| Method | Inc-v3 | Inc-v4 | Inc-Res-v2 | DN-121 | GoogleNet | VGG-16 |
|---|---|---|---|---|---|---|
| CGNC | 53.39±2.77 | 51.53±2.52 | 34.24±1.72 | 85.66±1.53 | 62.23±2.20 | 63.36±2.95 |
| Dual-Flow | 69.58±3.70 | 71.92±3.27 | 56.10±3.31 | 85.73±1.57 | 73.65±2.83 | 67.59±2.28 |

# M  Societal Impacts

Our work contributes to a deeper understanding of adversarial vulnerabilities in deep learning models, which is essential for improving model robustness and security. By developing more effective attack methods, we provide valuable tools for evaluating and strengthening defenses against malicious exploitation. However, as with any adversarial technique, there is a potential risk of misuse in compromising AI systems. We advocate for responsible use of such methods strictly within research and security auditing contexts, and encourage the community to develop corresponding mitigation strategies to safeguard AI applications.

