# OpenReview forum: "Dual-Flow: Transferable Multi-Target, Instance-Agnostic Attacks via $\textit{In-the-wild}$ Cascading Flow Optimization"
_NeurIPS.cc/2025/Conference — NeurIPS 2025 poster_

### Official Review · Reviewer_yoB3 · 2025-06-29

**Clarity:** 3
**Significance:** 3
**Originality:** 3
**Rating:** 4
**Confidence:** 1

**Summary:**

The paper introduces Dual-Flow, a novel framework for generating transferable, multi-target adversarial attacks in black-box settings. Unlike existing methods that struggle with low transferability or high computational costs, Dual-Flow leverages a two-stage flow-based approach: 1) Forward Flow: Uses a pretrained diffusion model to create an intermediate perturbed distribution; 2) Reverse Flow: Fine-tunes a LoRA-based velocity function for targeted adversarial refinement.

**Questions:**

- Are there any plans to extend the Dual-Flow framework to other domains or types of data beyond the current focus on image classification?
- Besides the attack success rate, could the authors consider including additional metrics (e.g., precision, recall) to provide a more comprehensive evaluation of the method's performance?

**Ethical Concerns:**

["NO or VERY MINOR ethics concerns only"]

**Final Justification:**

Most of my concerns have been addressed, so I am willing to maintain my original positive score.

**Limitations:**

yes

**Quality:**

3

**Strengths And Weaknesses:**

Strengths:
- The introduction of the Dual-Flow framework is a significant advancement, combining diffusion models with adversarial velocity functions for improved attack performance. The dual-flow pipeline (forward + reverse) elegantly balances perturbation generality and specificity.
- Propositions and theorems (e.g., cascading improvement in adjoint timesteps) provide a solid foundation for the approach.
- The authors provide extensive experiments demonstrating the effectiveness of their method across multiple benchmarks, showing clear improvements in attack success rates.

Weaknesses:
- The Dual-Flow framework may be complex for practical implementation, which could limit its accessibility for practitioners in the field.
- The paper could benefit from a more detailed analysis of the computational costs associated with the proposed method compared to existing approaches.
- Focused solely on image classification. Extensions to other domains (e.g., object detection, NLP) are unexplored but noted as future work.

---

> ### Author Rebuttal · Authors · 2025-07-31
>
> Thank you for your thoughtful review and detailed feedback on our paper. Below, we address the key points you raised:
>
> **The Dual-Flow framework may be complex for practical implementation, which could limit its accessibility for practitioners in the field.**
> We appreciate you raising this important point. While the conceptual framework involves two flows, the practical implementation is designed to be modular and manageable. Our approach hinges on two main components:
>
> 1. A Pre-trained Diffusion Model: We leverage a publicly available, off-the-shelf diffusion model (Stable Diffusion), which practitioners do not need to train from scratch.
>
> 2. A Lightweight Adversarial Flow: The core of our training is fine-tuning a velocity function using Low-Rank Adaptation (LoRA). LoRA is specifically designed to be parameter-efficient, only requiring the training of a small number of additional parameters. This significantly reduces the complexity and resource requirements compared to training a full model.
>
>
> Once our LoRA is trained, the inference process is very simple. We can directly use the image-to-image pipeline from existing code repositories (such as `diffusers`) to complete the process. It can be understood as performing DDIM Inversion on the original image, or the process of first adding noise then denoising, to generate adversarial samples.
>
> To further improve accessibility, we will add a detailed implementation guide in the appendix of our revised manuscript, clarifying the steps and highlighting the framework's modularity.
>
> **The paper could benefit from a more detailed analysis of the computational costs associated with the proposed method compared to existing approaches.**
>
> Thank you very much for your suggestion on this! Here we tested the training time required for the baseline methods `C-GSP` and `CGNC` on multi-target attack tasks using the same RTX 3090 GPUs as ours. Specifically, we require one day of training on two RTX 3090 GPUs, corresponding to 48 GPU hours, while this number is 50 GPU hours for `C-GSP` and 250 GPU hours for `CGNC`. This demonstrates that our model achieves superior baseline performance while maintaining excellent training efficiency. We will include this content and more detailed analysis in the final version.
>
> **Focused solely on image classification. Extensions to other domains (e.g., object detection, NLP) are unexplored but noted as future work.**
>
> We thank the reviewer for these forward-looking comments and question. Indeed, our current work establishes the Dual-Flow framework's effectiveness within the domain of image classification, which is a standard and foundational benchmark for new adversarial attack methods.
>
> We are very optimistic about the framework's potential for broader applications. The core principle of Dual-Flow—using a general pretrained flow for initial perturbation and a specialized, fine-tuned reverse flow for task-specific optimization—is highly adaptable.
>
> - For object detection, the adversarial loss function in the reverse flow could be modified to suppress or alter bounding box predictions.
>
> - For semantic segmentation, the goal could be to force specific pixels to be misclassified.
>
> - For NLP, since it differs significantly from our domain, we will continue to explore in the future how to apply similar ideas to related tasks.
>
> These are exciting avenues for future research that we intend to explore. We will emphasize these potential extensions more explicitly in the discussion and conclusion of the revised paper.
>
> **Are there any plans to extend the Dual-Flow framework to other domains or types of data beyond the current focus on image classification?**
>
> See the previous content.
>
> **Besides the attack success rate, could the authors consider including additional metrics (e.g., precision, recall) to provide a more comprehensive evaluation of the method's performance?**
>
> - We appreciate the suggestion to provide a comprehensive evaluation. For the specific task of targeted adversarial attacks, the Attack Success Rate (ASR) is the most direct and universally adopted metric in the literature. Additionally, we have also included 95% confidence intervals in the appendix for verification.
>
> - The goal of a targeted attack is singular: for a given input image, to mislead a model into predicting one specific, pre-determined target class. Therefore, the outcome for each attack is binary: it either succeeds (the model outputs the target class) or it fails. The ASR, which is the percentage of successful attempts over all tests, precisely captures the efficacy of the method in achieving this goal.
> - Metrics like precision and recall are standard for evaluating the performance of a classifier across a dataset with a ground-truth distribution of multiple classes. However, in our evaluation setting, we are not evaluating a classifier but the success of an attack. There is no concept of "true positives," "false positives," or "false negatives" in the traditional sense; there is only "attack success" or "attack failure" for the chosen target. Using ASR ensures our results are directly and fairly comparable to the established body of work on this topic. We will add a sentence in the experimental setup to clarify the rationale for using ASR as the primary metric in the context of targeted attacks.
> - We will continue to explore adding more metrics in the future to more comprehensively evaluate model performance. Thank you for your suggestion!
>
>
> We value your insights and believe they will significantly enhance our work's clarity and impact.

---

> > ### Comment · Reviewer_yoB3 · 2025-08-02
> >
> > Thank you for your response. Most of my concerns have been addressed, so I am willing to maintain my original positive score.

---

### Official Review · Reviewer_kcN4 · 2025-07-02

**Clarity:** 2
**Significance:** 2
**Originality:** 3
**Rating:** 4
**Confidence:** 5

**Summary:**

This paper introduces Dual-Flow framework for multi-target instance-agnostic adversarial attacks, utilizing Cascading Distribution Shift Training to develop an adversarial velocity function.

**Questions:**

The questions are the same as those mentioned in the weaknesses.

**Ethical Concerns:**

["NO or VERY MINOR ethics concerns only"]

**Final Justification:**

Most of my concerns have been addressed. Therefore, I would like to raise my rating. The authors are suggested to include their rebuttal responses and the additional experiments therein in their revised version.

**Limitations:**

Please see the weaknesses.

**Paper Formatting Concerns:**

There are no major formatting issues in this paper.

**Quality:**

2

**Strengths And Weaknesses:**

Strengths:

The topic of this paper is both interesting and important. The combination of a frozen diffusion ODE with a learnable reverse flow is conceptually appealing.


Weaknesses:

1. The selection of victim models is not clearly explained. It is unclear whether the victim models used in this paper are the same as those adopted in other baseline methods. The authors do not report the clean accuracy of the original classifiers, whether adversarially trained or not. In addition, the training details and procedures of these models are not provided and should be made transparent. Similarly, the selection of robust models and their relevant configurations should also be clearly stated.

2. The baseline methods selected by the authors are not very recent. For example, the CGNC method used for multi-target generative attacks is from 2023. The authors should include more recent baseline methods from the past two years for a fair and up-to-date comparison.

3. The baseline methods reported in Table 2 are fewer than those in Table 1. However, many methods applicable to multi-target attacks can also be adapted to the single-target setting. The authors should align the methods used in both tables for consistency. Additionally, the evaluation of the proposed method on adversarially trained models in the single-target setting is missing and should be included.

4. Only two input preprocessing defense methods (Gaussian smoothing and JPEG compression) are considered in the main paper. The authors should include more recent and advanced defense methods published in the past two years for a more comprehensive comparison.

5. The training efficiency of the proposed method is not reported. There is no comparison in training time or parameter count with baselines such as C-GSP or CGNC. Given that the multi-target model requires one day of training on dual RTX 3090 GPUs, such a comparison is necessary to evaluate cost-effectiveness.

6. The selection criteria for target classes in multi-target and single-target black-box attacks are unclear. The authors should clarify whether these classes were randomly selected or chosen based on vulnerability. In addition, the results only report mean success rates; standard deviation or confidence intervals should be provided to reflect the method’s stability.

7. The paper lacks more comprehensive ablation studies. For example, the number of target classes and the sampling steps should be ablated. Moreover, the effect of different perturbation budgets (e.g., 4/255, 8/255) is not explored. Without this, it is difficult to evaluate performance under low-budget attack constraints. In addition, the ablation experiments should include comparisons with more baseline methods. For example, in Figure 3, only the CGNC method is used for comparison, but other effective baselines should also be included.

8. The authors do not provide a theoretical explanation of why the proposed Dual-Flow sampling is effective in generating adversarial examples. A theoretical analysis should be included to demonstrate why adversarial examples generated in this way exhibit better transferability in black-box settings.

---

> ### Author Rebuttal · Authors · 2025-07-31
>
> Thank you for your thoughtful review and detailed feedback on our paper. Below, we address the key points you raised:
>
> **1. Victim Models**
> - **We completely follow the settings and models of baseline methods, maintaining complete consistency with them**. For the victim models we used, all models employ the **official weights** provided for the 1000-class ImageNet dataset. For normally trained models, we directly call these models and their weights through the `torchvision` or `timm` libraries. For robust models (adversarially trained models): `Inc-v3-adv` and `IR-v2-ens`, we call their official weights through the `timm` library; for `Res50-SIN`, `Res50-IN`, and `Res50-fine`, we use the weights provided by the open-source project `Texture-vs-Shape`; for `Res50-Aug`, we use the official weights provided by the open-source repository `AugMix`.
> - Since all the classifiers we use are pre-trained with official weights rather than training these models from scratch ourselves, the training details and accuracy information of these models are not reported by us. Thank you for your suggestion, we will supplement these details as much as possible in the final version!
>
> **2. Baseline Methods**
> - Thank you for your suggestion regarding the timeliness of baseline methods! We would like to point out that the CGNC method you mentioned was published at ECCV 2024. Unfortunately, for generative Instance-Agnostic Attacks methods, as of our paper submission deadline, we were unable to find more recent comparable open-source methods.
>
>
> **3. Content Reported in Tables**
> - Thank you for your careful observation! However, we must point out that those instance-specific attacks do appear in Table 1 but not in Table 2, but this is actually because for these methods, there is no distinction between multi-target attacks or single-target attacks, since their attacks do not involve training a model. In other words, these methods in Table 2 would only have exactly the same values as in Table 1. We did not show these methods in Table 2 for space-saving considerations.
> - Taking Res-152 as the surrogate model as an example, we present the complete table here.
>
>
>
> | Method | Inc-v3 | Inc-v4 | Inc-Res-v2 | Res-152 | DN-121 | GoogleNet | VGG-16 | Average |
> |:---:|:---:|:---:|:---:|:---:|:---:|:---:|:---:|:---:|
> | MIM | 0.50 | 0.40 | 0.60 | 99.70* | 0.30 | 0.30 | 0.20 | 0.38 |
> | TI-MIM | 0.30 | 0.30 | 0.30 | 96.50* | 0.30 | 0.40 | 0.30 | 0.32 |
> | SI-MIM | 1.30 | 1.20 | 1.60 | 99.50* | 1.00 | 1.40 | 0.70 | 1.20 |
> | DIM | 2.30 | 2.20 | 3.00 | 92.30* | 0.20 | 0.80 | 0.70 | 1.53 |
> | TI-DIM | 0.80 | 0.70 | 1.00 | 90.60* | 0.60 | 0.80 | 0.50 | 0.73 |
> | SI-DIM | 4.20 | 4.80 | 5.40 | 90.50* | 4.20 | 3.60 | 2.00 | 4.03 |
> | Logit | 10.10 | 10.70 | 12.80 | 95.70* | 12.70 | 3.70 | 9.20 | 9.87 |
> | SU | 12.36 | 11.31 | 16.16 | 95.08* | 16.13 | 6.55 | 14.28 | 12.80 |
> | GAP | 30.99 | 31.43 | 20.48 | 84.86* | 58.35 | 29.89 | 39.70 | 35.14 |
> | CD-AP | 33.30 | 43.70 | 42.70 | 96.60* | 53.80 | 36.60 | 34.10 | 40.70 |
> | TTP | 62.03 | 49.20 | 38.70 | 95.12* | 82.96 | 65.09 | 62.82 | 60.13 |
> | DGTA-PI | 66.83 | 53.62 | 47.61 | 96.48* | 86.61 | 68.29 | 69.58 | 65.42 |
> | CGNC$^{\dagger}$ | 68.86 | 69.45 | 45.71 | 98.61* | **91.14** | 69.83 | 68.05 | 68.84 |
> | Dual-Flow | 69.58 | 71.92 | 56.10 | 92.39* | 85.73 | 73.65 | 67.59 | 70.76 |
> | Dual-Flow$^{\dagger}$ | **72.25** | **74.35** | **58.44** | 93.65* | 87.61 | **75.45** | **71.11** | **76.12** |
>
> - Regarding the evaluation of our method on adversarially trained models in the single-target setting, we did not include this content due to considerations of the overall paper length and focus. Thank you for your suggestion, we will include this content in the final version!
>
>
> **4. Input Preprocessing Defense**
> - Thank you for your suggestion on this! We included Gaussian smoothing and JPEG compression in the main text, and included experiments on DiffPure in the appendix.  The experimental results are presented in the table below, where we show the attack success rates of our method under various DiffPure $t^*$ settings and compare them with baseline methods (using ResNet-152 as the white-box model). As illustrated in the table, baseline methods are easily nullified by the purification process, whereas our method maintains a significant success rate. This further demonstrates the robustness of our approach.
>
> | $t^*$ | Method    | Inc-v3 | Inc-v4 | Inc-Res-v2 | Res-152 | DN-121 | GoogleNet | VGG-16  |
> |:----:|:---------:|:------:|:------:|:----------:|:-------:|:------:|:---------:|:-------:|
> | 0.05 | CGNC      | 16.26  | 19.91  | 8.53       | 67.76   | 49.81  | 21.79     | 29.81   |
> | 0.05 | Dual-Flow | 49.60  | 51.92  | 37.56      | 79.50   | 70.20  | 51.30     | 52.26   |
> | 0.1  | CGNC      | 2.41   | 2.96   | 1.25       | 14.65   | 10.10  | 3.46      | 4.59    |
> | 0.1  | Dual-Flow | 24.31  | 25.32  | 18.76      | 48.05   | 39.81  | 25.06     | 24.78   |
> | 0.15 | CGNC      |  0.47  |  0.46  |  0.34      | 1.84    | 1.46   |   0.62    | 0.92    |
> | 0.15 | Dual-Flow |  7.20  | 7.87   |  5.89      | 16.70   | 13.72  |   7.71    |  8.34   |
>
> - We will continue to add updated defense methods in the final version to further improve the comprehensiveness of our comparison.
>
> **5. Training Efficiency**
> - Thank you very much for your suggestion on this! Here we tested the training time required for the baseline methods `C-GSP` and `CGNC` on multi-target attack tasks using the same RTX 3090 GPUs as ours. Specifically, we require one day of training on two RTX 3090 GPUs, corresponding to 48 GPU hours, while this number is 50 GPU hours for `C-GSP` and 250 GPU hours for `CGNC`. This demonstrates that our model achieves superior baseline performance while maintaining excellent training efficiency. We will include this content and more detailed analysis in the final version.
>
> **6. Target Classes Selection and Method Stability Reporting**
>
> - Regarding the selection of target classes, to facilitate comparison with baselines, we follow previous work that used the NeurIPS 2017 adversarial competition dataset. Consistent with these works, we select 8 fixed target classes: `150, 426, 843, 715, 952, 507, 590, 62`.
> - Thank you for your suggestion on method stability evaluation! We have provided relevant content in Appendix J. We computed 95% confidence intervals (CIs) for our method. As shown in the table below, the non-overlapping intervals observed in our experiments indicate that the improvement in attack success rate achieved by our method is statistically significant with high confidence.
>
>
> | Method | Inc-v3 | Inc-v4 | Inc-Res-v2 | DN-121 | GoogleNet | VGG-16 |
> |:---:|:---:|:---:|:---:|:---:|:---:|:---:|
> | CGNC | 53.39±2.77 | 51.53±2.52 | 34.24±1.72 | 85.66±1.53 | 62.23±2.20 | 63.36±2.95 |
> | Dual-Flow | 69.58±3.70 | 71.92±3.27 | 56.10±3.31 | 85.73±1.57 | 73.65±2.83 | 67.59±2.28 |
>
> **7. Ablation Studies**
> Thank you for your suggestion regarding more comprehensive ablation studies! In addition to the main text, we provide some additional ablation experiments in the appendix (supplementary materials). Regarding your points:
> - For the number of target classes, we will complete this experimental content in the final version according to your suggestion as much as possible.
> - For sampling steps, we already have some experiments in Figure 4. Of course, we can find ways to complete this content more thoroughly. Thank you for your suggestion!
> - For the effect of lower perturbation budgets. This is a very good suggestion! We chose $\epsilon=16/255$ following the default setting used in previous work on the NIPS 2017 dataset to ensure direct comparability. However, we agree that evaluating adversarial perturbations at smaller budgets is important for enhancing visual stealthiness.
> To address your suggestion, we conducted experiments at lower perturbation budgets for black-box targeted attacks. We used the $\epsilon=16/255$ versions of these models and clipped the generated samples to meet the $\epsilon=8/255$ and $\epsilon=4/255$ limits. For Logit and SU, we regenerated adversarial perturbations under the new budgets.
> The results are shown below. Values before the slash represent $\epsilon=8/255$, and values after the slash represent $\epsilon=4/255$. Our method demonstrates superior black-box transferability compared to baseline methods.
>
> Method|Inc-v4|Inc-Res-v2|Res-152|DN-121
> :---:|:---:|:---:|:---:|:---:
> Logit|1.40/0.50|1.01/0.28|0.90/0.24|0.90/0.26
> SU|1.56/0.30|1.01/0.26|0.93/0.28|1.00/0.31
> CGNC|13.33/0.29|6.54/0.07|6.95/0.16|14.65/0.26
> Dual-Flow|23.86/2.74|14.04/1.18|21.21/1.46|27.68/2.18
>
> - For the suggestion of ablation experiments with more baseline methods. We will include this content in the final version as much as possible.
>
> Thank you again for your detailed suggestions!
>
> **8. Theoretical Explanation**
>
> Thank you for your suggestion! We will further include more detailed theoretical analysis in the final version. We have some explanations in Appendix G. Additionally, we believe that the generator constrains feasible perturbations to a low-dimensional manifold (determined by network parameters), which acts as a form of strong regularization. This makes it easier to find "wide" and stable adversarial regions rather than narrow edge solutions, making it less sensitive to small changes. This helps reduce dependence on any single decision boundary shape (i.e., on the surrogate model) and improves success rates under defenses/randomization. We will add more theoretical analysis content as much as possible in the future, thank you again!
>
> We value your insights and believe they will significantly enhance our work's clarity and impact.

---

> > ### Comment · Reviewer_kcN4 · 2025-08-07
> >
> > Thank you for offering detailed responses! Most of my concerns have been addressed. Therefore, I would like to raise my rating.

---

> ### Comment · Area_Chair_xtRm · 2025-08-06
>
> Dear reviewer,
>
> Does the response address your concerns? Could you please give a response?
>
> Best,
>
> AC

---

### Official Review · Reviewer_yjD2 · 2025-07-02

**Clarity:** 3
**Significance:** 3
**Originality:** 3
**Rating:** 5
**Confidence:** 3

**Summary:**

The paper tackles the problem of transferable multi-target instance-agnostic adversarial attack. To do so, it leverages the generative power of diffusion models via a Dual-flow framework. Specifically, it combines a pre-trained diffusion model and a LoRA-based adversarial velocity function. It also proposes the Cascading Distribution Shift Training, which is designed to improve the efficacy of adversarial attack. The proposed method outperforms existing multi-target adversarial attacks in diverse settings.

**Questions:**

N/A

**Ethical Concerns:**

["NO or VERY MINOR ethics concerns only"]

**Final Justification:**

My original concerns were on the validity of the proposed method under more difficult attack settings (i.e., lower perturbation bounds). The results provided in the rebuttal has shown the superiority of the proposed method under more difficult settings, which addresses my concern.

**Limitations:**

Yes.

**Quality:**

3

**Strengths And Weaknesses:**

Strengths:
1. The paper proposes a novel concept of applying flow-based ODE velocity for adversarial attacks.
2. The paper achieves state-of-the-art performance on multi-target adversarial attack.
3. The paper provides an extensive number of empirical evaluations on the proposed attack.

Weaknesses:
1. The visualizations in Fig. 2 show that the perturbations are highly visible. Also, these perturbations seem to be semantically structured and resemble features of the target class. This suggests that the attack may rely on large, structured perturbations rather than subtle changes. The paper needs more visualization and evaluation on smaller perturbation budgets where such semantic structures would be less visible.
2. The paper lacks evaluation on transferability across CNN-based and transformer-based models, which are more popular recently. How does the transferability of the attack perform across these models with larger discrepancies?

---

> ### Author Rebuttal · Authors · 2025-07-31
>
> Thank you for your thoughtful review and detailed feedback on our paper. Below, we address the key points you raised:
>
> ## Evaluation on Smaller Perturbation Budgets
> - Thank you for your suggestion. Unfortunately, due to NeurIPS mechanisms this time, we are unable to further demonstrate visualization effects to you. We have conducted such tests: for different original images, we input the same target class prompt and observe the resulting adversarial perturbations. It can be seen that the perturbations are highly dependent on the structure and layout of the original image, rather than being solely determined by the target class prompt. In other words, the mechanism by which our attack works is not simply adding target class structure and semantic information to the original image.
> - We chose $\epsilon=16/255$ following the default setting used in previous work on the NIPS 2017 dataset to ensure direct comparability. However, we agree that evaluating adversarial perturbations at smaller budgets is important for enhancing visual stealthiness.
> To address your suggestion, we conducted experiments at lower perturbation budgets for black-box targeted attacks. We used the $\epsilon=16/255$ versions of these models and clipped the generated samples to meet the $\epsilon=8/255$ and $\epsilon=4/255$ limits. For Logit and SU, we regenerated adversarial perturbations under the new budgets.
> The results are shown below. Values before the slash represent $\epsilon=8/255$, and values after the slash represent $\epsilon=4/255$. Our method demonstrates superior black-box transferability compared to baseline methods.
>
> Method|Inc-v4|Inc-Res-v2|Res-152|DN-121
> :---:|:---:|:---:|:---:|:---:
> Logit|1.40/0.50|1.01/0.28|0.90/0.24|0.90/0.26
> SU|1.56/0.30|1.01/0.26|0.93/0.28|1.00/0.31
> CGNC|13.33/0.29|6.54/0.07|6.95/0.16|14.65/0.26
> Dual-Flow|23.86/2.74|14.04/1.18|21.21/1.46|27.68/2.18
>
> Reducing the perturbation budget makes adversarial attacks more difficult, especially for combined black-box and target attacks, which remains a challenge in adversarial research. Enhancing attack effectiveness under subtle conditions is valuable. We plan to explore this further to develop techniques that remain effective while being less perceptible to the human eye.
>
> ## Transferability across CNN-based and Transformer-based Models
> As you correctly pointed out, it is important to validate the transferability of our method across models with significantly different architectures. We have already provided this content in the appendix submitted as supplementary materials, specifically in Appendix E. We tested the attack success rates from Res-152 to various Transformer architecture models. We have reproduced these data in the table below, which demonstrates that our method remains effective for transfer between models with substantially different architectures.
>
> | Method | ViT-B/16 | CaiT-S/24 | Visformer-S | DeiT-B | LeViT-256 | TNT-S |
> |:----------:|:------------:|:-------------:|:---------------:|:----------:|:-------------:|:----------:|
> | C-GSP      | 11.78        | 32.00          | 36.60            | 35.58      | 37.85         | 31.00         |
> | CGNC       | 19.46        | 54.56         | 58.70            | 59.90       | 57.53         | 48.40       |
> | Dual-Flow  | **36.39**        | **74.24**         | **76.72**           | **78.50**       | **79.34**         | **67.86**      |
>
> We value your insights and believe they will significantly enhance our work's clarity and impact.

---

> > ### Comment · Reviewer_yjD2 · 2025-08-05
> > **Thank you for the rebuttal**
> >
> > I appreciate the authors for providing a rebuttal. My concerns have been addressed, and I raise my score to Weak Accept.

---

### Official Review · Reviewer_iNaw · 2025-07-06

**Clarity:** 3
**Significance:** 3
**Originality:** 3
**Rating:** 4
**Confidence:** 5

**Summary:**

This paper introduces Dual-Flow, a novel framework for generating multi-target, instance-agnostic adversarial attacks with high transferability in black-box settings. The method integrates a pretrained diffusion model (forward flow) with a trainable reverse flow, optimized via a new Cascading Distribution Shift Training algorithm. This reverse flow is designed to refine perturbations toward a target class using ODE-based integration. The method is empirically shown to outperform previous state-of-the-art attacks in both black-box transferability and robustness against defenses, particularly in multi-target settings.

**Questions:**

Please address the questions highlighted in the weaknesses.

**Ethical Concerns:**

["NO or VERY MINOR ethics concerns only"]

**Quality:**

3

**Strengths And Weaknesses:**

## Strengths

1. **Innovative Dual-Flow Architecture:**
   The separation of forward and reverse flows—leveraging pretrained diffusion and a fine-tuned LoRA-based velocity function—is conceptually elegant and practically effective for structured adversarial perturbation.

2. **Cascading Distribution Shift Training:**
   This novel optimization strategy addresses the difficulty of training with in-the-wild trajectories and enables progressive, semantically meaningful perturbation refinement.

3. **Strong Empirical Results:**
   The method demonstrates significant improvements (e.g., +34.58% success rate from Inc-v3 to ResNet-152) over existing multi-target generative attacks across multiple black-box and robust models.

## Weaknesses

1. **Limited Architectural Diversity in Transfer Evaluation:**
   - Two surrogate models, ResNet-152 and Inception-v3, are used in training, and all target models are CNN-based. The method’s effectiveness against architecturally diverse models (e.g., Vision Transformers) is not evaluated. If the target model differs significantly from the surrogate models, transferability may be substantially reduced.
   - There is no detailed ablation study comparing the individual versus combined contributions of the two surrogate models. This limits the generalizability of the claimed transferability.

2. **Training Cost and Practical Scalability:**
   While the Dual-Flow framework is positioned as a scalable alternative to single-target generative attacks, the actual training cost undermines this motivation. According to the paper:
   - Training the multi-target model (for 8 target classes) takes approximately 1 day on 2 RTX 3090 GPUs (about 48 GPU-hours).
   - In contrast, training a single-target model requires about 4 hours on 1 GPU, so training 8 single-target models would take roughly 32 GPU-hours in total.
   Thus, multi-target training is actually more expensive than training individual single-target models, despite the higher architectural complexity and parameter sharing across classes. This contradicts the presumed efficiency advantage of multi-target approaches and raises concerns about practical scalability, especially when extending to more target classes or higher input resolutions.
   Moreover, Table 2 shows that single-target fine-tuned variants (Dual-Flow†) actually outperform the multi-target model, indicating that the multi-target setup sacrifices both efficiency and performance for flexibility. The paper does not fully address or justify this trade-off.

3. **No Limitation Section:** The paper claims to discuss its limitations in the Appendix, but no such section or detailed discussion is provided.

4. **Terminology Confusion Regarding “Victim Model”:**
   - Algorithm 1 refers to the training model \( f \) as the "victim model," which is used to compute gradients.
   - However, in a true black-box setting, attackers do not have access to victim gradients.
   - This introduces confusion: in reality, \( f \) should be a surrogate model, not the actual target. This inconsistency should be clarified.

---

> ### Author Rebuttal · Authors · 2025-07-31
>
> Thank you for your thoughtful review and detailed feedback on our paper. Below, we address the key points you raised:
>
> ## Limited Architectural Diversity in Transfer Evaluation
> - As you correctly pointed out, it is important to validate the transferability of our method across models with significantly different architectures. We have already provided this content in the appendix submitted as supplementary materials, specifically in Appendix E. We tested the attack success rates from Res-152 to various Transformer architecture models. We have reproduced these data in the table below, which demonstrates that our method remains effective for transfer between models with substantially different architectures.
>
> | Method | ViT-B/16 | CaiT-S/24 | Visformer-S | DeiT-B | LeViT-256 | TNT-S |
> |:----------:|:------------:|:-------------:|:---------------:|:----------:|:-------------:|:----------:|
> | C-GSP      | 11.78        | 32.00          | 36.60            | 35.58      | 37.85         | 31.00         |
> | CGNC       | 19.46        | 54.56         | 58.70            | 59.90       | 57.53         | 48.40       |
> | Dual-Flow  | **36.39**        | **74.24**         | **76.72**           | **78.50**       | **79.34**         | **67.86**      |
>
> - Thank you for your suggestion, we will include detailed analysis and experiments for two different surrogate models in the final version. Currently, we know that transfer between structurally similar models is easier. For example, transfer from Inc-v3 to Inc-v4 is easier compared to Res-152. We will conduct further analysis in the future.
>
> ## Training Cost and Practical Scalability
> - Thank you for your consideration regarding efficiency. We apologize if our description may have caused misunderstanding. In fact, the single-target variants we proposed in the paper are obtained through **further fine-tuning** of models that were previously trained under multi-target settings. That is to say, to obtain single-target models for all 8 classes, the actual computational cost required is 48+32=80 GPU hours, rather than only 32 GPU hours. Therefore, it cannot be said that multi-target training is more expensive than individual target models.
> - This also explains why our single-target variants achieve higher attack success rates than multi-target models: if there were no performance improvement, the additional 32 GPU hours would be wasted. In other words, further fine-tuning on single targets is precisely to improve success rates. Of course, your concern about the trade-off between flexibility and performance is very meaningful. Regarding this aspect, we have some explanations in Section 4.2. Specifically, our model can still achieve performance exceeding baselines (which are all single-target models) even without further fine-tuning on individual target classes. The single-target variants are only for fair comparison settings.
> - Additionally, you can observe the performance improvement that the baseline (CGNC) achieves after single-target fine-tuning. It can be seen that the performance difference between our multi-target model and single-target model is smaller. This indicates that the impact of our method's flexibility on performance is relatively small.
>
> | Source | Dual-Flow | Dual-Flow$^{\dagger}$ | CGNC | CGNC$^{\dagger}$ |
> |:---:|:---:|:---:|:---:|:---:|
> |Inc-v3|73.96|76.79|52.80|70.00|
> |Res-152|70.76|76.12|58.40|68.84|
>
> - Additionally, our multi-target model requires 48 GPU hours for training. Through our actual testing on the same RTX 3090, the baseline CGNC requires 250 GPU hours for training. This also demonstrates that we have lower training overhead.
>
> ## No Limitation Section
> Thank you for your concern about the Limitation section of our paper. In fact, our appendix is provided in the supplementary materials, and the Limitation Section is located in Appendix I.
>
> ## Terminology Confusion Regarding "Victim Model"
> Thank you for your very important suggestion regarding our terminology. As you correctly pointed out, since our goal is black-box attacks, it is inappropriate to refer to ($f$) in Algorithm 1 as "victim model". We will modify it to "surrogate model" in the final version.
>
> We value your insights and believe they will significantly enhance our work's clarity and impact.

---

> > ### Comment · Reviewer_iNaw · 2025-08-08
> >
> > Why don't you compare single-target and multi-target approaches on a fair basis (i.e., without further fine-tuning for the former)? It doesn't make much sense to report results that cannot be compared fairly. Intuitively, I believe that single-target models should perform better than multi-target ones, since they are simpler and easier to train. In practice, higher transferability is a higher priority than training time for this attack, especially if the training times are not significantly different—particularly given that training is a one-time process.

---

> > > ### Author Response · Authors · 2025-08-08
> > > **Regarding Fair Comparison on Single-Target Tasks**
> > >
> > > Thank you very much for your valuable comments. We fully agree with your point that transferability is a higher priority than training time in this setting—if the attack is ineffective, efficiency becomes secondary.
> > >
> > > Regarding the fairness of the comparison, in **Table 2** of our paper, *Dual-Flow* denotes our multi-target model without any single-target fine-tuning, whereas *Dual-Flow†* is the single-target fine-tuned variant. In other words, the results of *Dual-Flow* already correspond to the “without further fine-tuning” condition you mentioned, and we directly compare them with other single-target baselines under this setting. As can be seen, even without any single-target fine-tuning, our multi-target *Dual-Flow* already achieves superior transferability over other baselines, which aligns with the fair comparison principle you suggested.
> > >
> > > The reason we additionally report single-target fine-tuned results (*Dual-Flow†*) is to follow the evaluation protocol adopted by previous works. For example, the baseline *CGNC†* is obtained by fine-tuning a multi-target model on a single target class, which can further boost its performance. For completeness and comparability, we also include this variant.
> > >
> > > That said, the main focus of our work is on the design and effectiveness of the multi-target model itself. We hope this clarification addresses your concern, and we would be very happy to continue the discussion if anything remains unclear.

---

### Decision · Program_Chairs · 2025-09-17

**Decision:**

Accept (poster)

**Comment:**

This paper proposes a novel framework named as Dual-Flow for generating multi-target, instance-agnostic adversarial attacks with high transferability in black-box settings. It combines a pretrained diffusion model (forward flow) with a trainable reverse flow, optimized via a new Cascading Distribution Shift Training method. Experiments demonstrates the effectiveness of their proposed method.

Strengths
- The technology of Dual-Flow Architecture is novel. The separation of a pretrained diffusion model (forward flow) and a LoRA-finetuned velocity function (reverse flow) is conceptually clean and effectively enables structured adversarial perturbations.

- The method significantly outperforms existing approaches on multiple black-box and robust models.

Weakness

- The method uses the diffusion model, which may bring large computational cost.

In overall, the paper is novel in its technology and effective in its results, overweigh its weakness. I recommend to its acceptance.